# Mitochondrial reactive oxygen species regulate the induction of CD8+ T cells by plasmacytoid dendritic cells

Marine Oberkampf[1,2], Camille Guillerey [1,2,7], Juliette Mouriès[1,2], Pierre Rosenbaum [1,2], Catherine Fayolle[1,2], Alexandre Bobard[3], Ariel Savina[4,5], Eric Ogier-Denis[6], Jost Enninga[3], Sebastian Amigorena[4], Claude Leclerc [1,2] & Gilles Dadaglio[1,2]

Cross-presentation allows exogenous antigen presentation in association with major histocompatibility complex class I molecules, a process crucial for the priming of CD8+ T-cell responses against viruses and tumors. By contrast to conventional dendritic cells (cDC), which cross-present antigens in the steady state, plasmacytoid dendritic cells (pDC) acquire this ability only after stimulation by Toll-like receptor (TLR) ligands. The intracellular pathways accounting for this functional difference are still unknown. Here we show that the induction of cross-presentation by pDCs is regulated by mitochondria through a reactive oxygen species (ROS)-dependent mechanism, involving pH alkalization and antigen protection. The reduction of mitochondrial ROS production dramatically decreases the cross-presentation capacity of pDCs, leading to a strong reduction of their capacity to trigger CD8+ T-cell responses. Our results demonstrate the importance of mitochondrial metabolism in pDC biology, particularly for the induction of adaptive immune responses.

[1] Institut Pasteur, Unité de Régulation Immunitaire et Vaccinologie, Equipe Labellisée Ligue Contre le Cancer, 75015 Paris, France. [2] INSERM U1041, 75015 Paris, France. [3] Département de Biologie Cellulaire et Infection, Institut Pasteur, Dynamique des Interactions Hôte-Pathogène, 75015 Paris, France. [4] INSERM U932, Institut Curie, 75248 Paris, France. [5] Roche SAS, 92650 Boulogne-Billancourt Cedex, France. [6] INSERM U773-CRB3, Faculté de Médecine Xavier Bichat, 75018 Paris, France. [7] Present address: Immunology of Cancer and Infection Laboratory, QIMR Berghofer Medical Research Institute, Herston, 4006 Queensland, Australia. These authors contributed equally: Marine Oberkampf, Camille Guillerey, Juliette Mouriès. These authors jointly supervised this work: Claude Leclerc, Gilles Dadaglio. Correspondence and requests for materials should be addressed to C.L. (email: claude.leclerc@pasteur.fr) or to G.D. (email: gilles.dadaglio@pasteur.fr)

Cross-presentation allows exogenous antigen (Ag) presentation in association with MHC class I molecules. Cross-presentation is performed most efficiently by dendritic cells (DCs) and is crucial for the development of CD8[+] T-cell responses against tumors and viruses that do not infect antigen-presenting cells (APCs). Among all APCs, conventional dendritic cells (cDC1) (i.e. mouse CD8α[+] DCs and their human counterpart, BDCA3[+] DCs) have the unique property of cross-presenting exogenous Ags constitutively[1–3]. These APCs are endowed with a specialized phagocytic pathway that allows them to perform an efficient cross-presentation[4,5]. After their capture by endocytosis or phagocytosis, Ags are transferred from the endocytic compartments to the cytosol where they are degraded by the proteasome into 8–9 amino acid peptides. These processed Ags are then transported by TAP either to the ER or endosomes to be loaded onto MHC class I molecules. To be efficiently processed, exogenous Ags must be protected from extensive degradation in the endocytic compartment to prevent the destruction of potential T-cell epitopes. Reactive oxygen species (ROS) are involved in this process[6]. ROS are small molecules produced by living organisms through the partial reduction of oxygen[7] and play an important role in physiological cell functions and immune regulation[8]. ROS are mainly produced by NADPH oxidase (NOX) complexes or an electron leak from mitochondrial aerobic respiration. The specialized phagocytic pathway of cDC1s includes the recruitment of NOX2 to the early phagosome by a Rab27a-dependent process, which mediates the sustained production of low levels of ROS. This ROS production leads to the alkalization of phagosomal pH through the consumption of protons, which prevents Ag degradation by the inhibition of the acidic lysosomal proteases. This process allows for the efficient proteasome-mediated processing of exogenous Ags after their transport from the lumen of the endocytic compartments to the cytosol[6,9]. The cross-presentation by cDC1s could be enhanced by TLR engagement, which induces the inhibition of phago-lysosome fusion by a Rab34-dependent mechanism that delays Ag degradation[10].

Nevertheless, while cDC1s constitute the most efficient cross-presenting cells, other DC subsets can also induce the priming of CD8[+] T cells against exogenous Ags, particularly after activation[11–14].

Notably, plasmacytoid DCs (pDCs) are endowed with Ag-presenting cell functions[15], and several groups have reported that in both humans and mice, pDCs have the capacity to present exogenous Ags to CD8[+] T cells[12,14,16,17]. Compared with cDC1s, which prime T cells in secondary organs, pDCs might play a crucial role in T-cell activation at the site of inflammation[15]. However, resting mouse splenic pDCs are unable to cross-present Ags and acquire this capacity only following TLR stimulation. Thus, unlike cDC1s, the pDC cross-presentation ability is tightly regulated and depends on their activation state[16]. The intracellular pathways that support cross-presentation in pDCs, however, remain largely unexplored.

In the present study, we examine the role of ROS in the cross-presentation of exogenous Ags by pDCs. Following the activation of pDCs by TLR-L, which induces cross-presentation in pDCs, increased production of ROS is observed and is associated with a high endosomal pH, Ag protection from endosomal degradation and export to the cytosol, which is consistent with previously demonstrated observations in cDC1s. However, unlike cDC1s, the induction of cross-presentation in pDCs is independent of NOX2. Instead, activation of pDCs by TLR ligation induces production of mitochondrial ROS (mROS). Using transgenic mice that express the human catalase, which reduces $H_2O_2$ production by the mitochondria, we show that mROS play a crucial role in the induction of cross-presentation in pDCs. Importantly, we

demonstrate that reduced mROS levels in mitochondria affect the generation of CD8[+] T-cell responses in vivo. Therefore, the induction of cross-presentation in pDCs, and their ability to cross prime CD8[+] T-cell responses, are both dependent on mitochondrial ROS production. These results uncover a new, unexpected, function for mitochondrial respiration in the control of endocytic functions and the induction of adaptive immune responses.

## Results

**Activated pDCs cross-present different exogenous Ags.** In a previous study, using exogenous, soluble and particulate Ags, we demonstrated that the capacity of murine pDCs to cross-present Ags is not constitutive but can be induced following stimulation by TLR7 or 9 ligands[16]. Ag internalization constitutes the first step of cross-presentation, and, thus, the cross-presenting capacity of pDCs could vary according to the source of the Ags. Apoptotic cells constitute the main source of the exogenous Ags that are presented in vivo by MHC-I molecules[18]. We, thus, first analyzed the capacity of pDC and cDC subsets to capture various types of exogenous Ags, such as proteins, particulate and cell-associated Ags (Supplementary Fig. 1a, b). Despite differences in efficiency, all DC subsets were able to internalize these various Ags. In addition, imaging flow cytometry showed that captured apoptotic cells were located inside the pDCs and not at the cell surface, thus demonstrating that these cells have internalized the dying cells (Supplementary Fig. 1c). Notably, the pDC activation did not significantly affect their capacity to capture soluble Ags (Supplementary Fig. 1d). We then tested the ability of H-2[b] pDCs to cross-present different formulations of exogenous Ags, particularly Ags from allogenic or H-2K[b−/−] apoptotic cells. Purified pDCs were loaded with different sources of OVA in the medium or with R848 and were then co-cultured with OVA-specific OT-I cells (Fig. 1a). While no T-cell proliferation was observed with the resting pDCs, the R848-activated pDCs loaded with OVA, the OVA-coated beads or the OVA-associated apoptotic splenocytes induced a strong T-cell response. In this experiment, the OVA-loaded or mOVA apoptotic cells could not directly present the OVA Ag to T cells since they were isolated from the MHC I mismatch (K[d]) BALB/c or K[b−/−] mice, respectively. These data demonstrate that R848-activated pDCs process different sources of exogenous Ags and present them to CD8[+] T cells.

**ROS neutralization inhibits cross-presentation by pDCs.** It has been demonstrated that NOX2-dependent ROS production is involved in the cross-presentation ability of cDC1s[19]. Thus, to assess whether ROS also regulate the pDC ability to cross-present Ags, we treated pDCs with either DPI, which is an inhibitor of flavoenzymes, such as NOXs[20], or NAC, which is an ROS scavenger[21]. Both DPI and NAC fully inhibited the cross-presentation of soluble OVA by activated pDCs in a dose-dependent manner (Fig. 1b, c) in the absence of toxicity (Supplementary Fig. 2a). Similar results were obtained with the OVA-coated beads and the apoptotic OVA-expressing cells Ag (Supplementary Fig. 2b), while the presentation of the SIINFEKL-peptide was not affected by DPI or NAC (Fig. 1b, c and Supplementary Fig. 2b). In addition, we found that apocynin and plumbagin, which are two other antioxidant molecules[20,22], strongly reduced the pDC ability to cross-prime OT-I cells (Supplementary Fig. 2c). In contrast, while autophagy was demonstrated to be involved in cross-presentation by DCs[23], autophagy was not involved in the cross-presentation by pDCs following R848 activation since the pDCs that were isolated from the autophagy-deficient GCN2 mice cross-presented exogenous

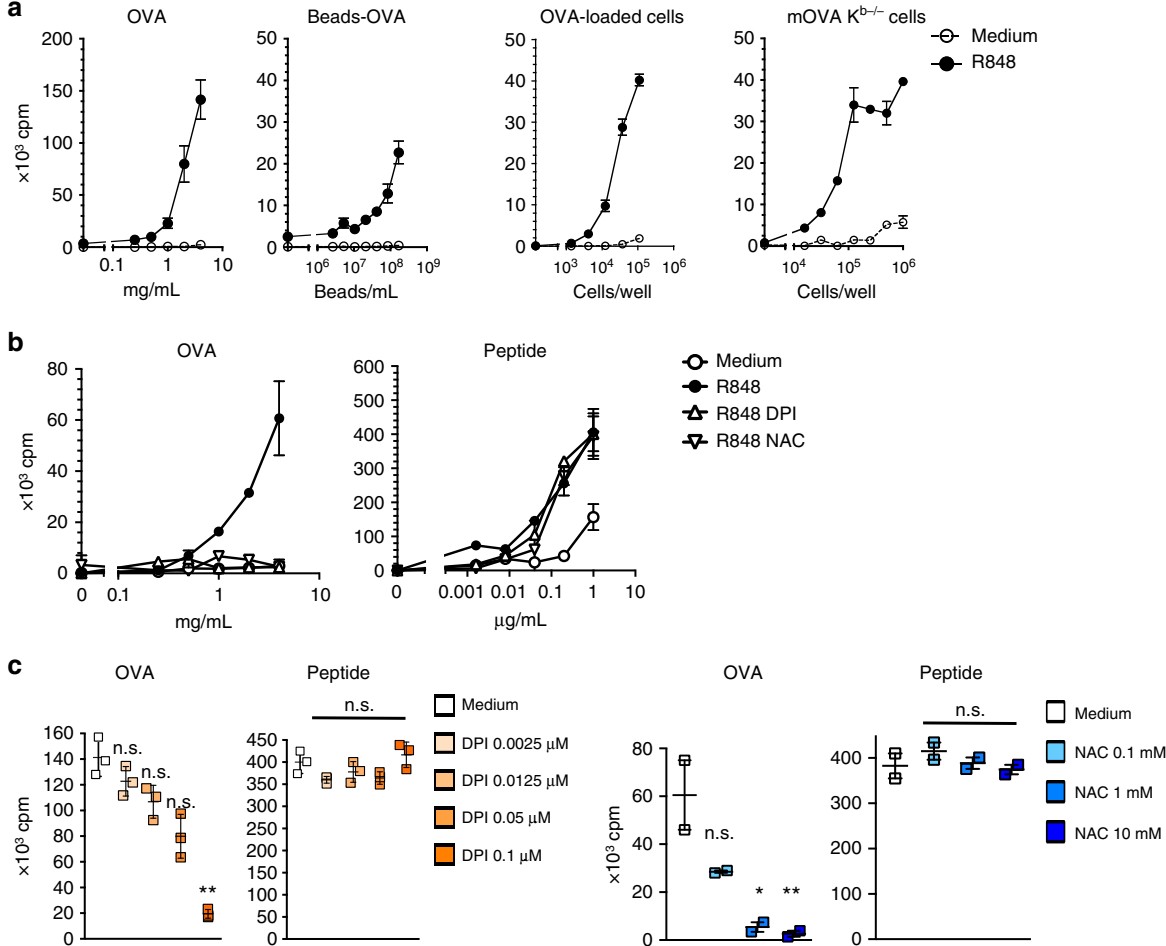

**Fig. 1** TLR7-induced cross-presentation by pDCs is inhibited by the ROS inhibitor and scavenger. $10^4$ purified pDCs from C57BL/6 mice were incubated for 1 h with different Ags in the presence or absence of R848 (1 μg/mL) and inhibitors. **a** pDCs were incubated together with serial dilutions of the OVA protein (OVA), OVA-coated beads (Beads-OVA), apoptotic OVA-loaded BALB/c ($K^d$) splenocytes (OVA-loaded cells) or apoptotic OVA-expressing $K^b$$^{-/-}$ splenocytes (mOVA cells) from Act-mOVA/ $Kb^{-/-}$ mice in the medium alone or with R848. **b** pDCs were incubated with serial dilutions of OVA or the SIINFEKL OVA peptide in the medium with R848 alone or with DPI (0.1 μg/mL) or NAC (10 mM). **c** pDCs were incubated with the OVA protein (1 mg/mL) or the SIINFEKL OVA peptide (0.001 μg/mL) and R848 alone or with serial dilutions of DPI or NAC. **a**−**c** After washing, the pDCs were co-cultured with LN cells from the OT-I Rag$^{-/-}$ mice for 72 h. The T-cell proliferation is expressed in cpm of incorporated [$^3$H]-thymidine. The results are shown as the mean cpm ± SD of triplicates (**a**−**c**) or duplicates (**c**) and each dot represents one replicate. One representative experiment of 3 (**a**, **b**) or 2 (**c**) is depicted. Significant differences were analyzed for each condition compared to the medium alone with an unpaired $t$ test; n.s., non-significant, *$p < 0.05$; **$p < 0.01$

Ags as efficiently as the pDCs from C57BL/6 mice (Supplementary Fig. 3).

In agreement with our previous study[16], the capacity of pDCs to cross-present Ags was also induced following CpG activation (Supplementary Fig. 4a, b). This induction was strongly reduced by ROS inhibitors, suggesting that the induction of cross-presentation by R848 and CpG is regulated by similar mechanism(s). Altogether, these data show that ROS are required for the acquisition of the cross-presenting ability by pDCs, following TLR-L activation.

**TLR-L stimulation triggers cytosolic ROS production in pDCs**. The inhibition of the cross-presentation of the activated pDCs by the ROS inhibitor and scavenger suggests that the activation of pDCs induces ROS production. We, thus, quantified the ROS production in TLR-L-activated pDCs using 2′,7′-dichloro-fluorescin diacetate (DCFDA), which is a fluorescent biosensor that emits light at 530 nm when oxidized. Low levels of fluorescence were detected at 530 nm in the resting pDCs, indicating

that low amounts of ROS were present in the cytosol at the steady state (Fig. 2a). After activation by R848 (Fig. 2a) or CpG (Supplementary Fig. 4c), a strong ROS production was detected. This TLR-L-induced ROS production by pDCs was inhibited by the DPI and NAC treatments (Fig. 2b, c).

To gain further insight into the mechanisms regulating the cytosolic ROS concentrations in pDCs following R848 activation, we analyzed the expression of 86 genes that are involved in oxidative stress and antioxidant defense. Using the software Qlucore for an initial analysis, we found that only seven genes were regulated upon activation when the $q$ score was <0.1 (Fig. 2d). The expression of glutathione peroxidases (Gpx) 1 and 6, peroxiredoxins (Prdx) 2 and 5 and thioredoxin interacting protein (Txnip) was reduced, whereas the expression of thioredoxin reductase 1 (Txnrd1) and vimentin (Vim) was increased.

To ensure that we did not exclude any genes by choosing an arbitrary cutoff, we performed a second statistical analysis in which each gene was monitored individually. This approach led to the identification of 18 genes that are modified by the R848

treatment, including the 7 previously identified genes (Fig. 2e). In total, 15 genes were downregulated, and 3 genes were upregulated by the activation. In addition to Gpx1 and 6 and Prdx2 and 5, we observed a downregulation in the expression of Prdx3 and catalase. Interestingly, glutathione peroxidases, peroxiredoxins, and catalase are $H_2O_2$ degrading enzymes. Hence, a decrease in their expression could leave cells exposed to high levels of $H_2O_2$.

These data suggest that the regulation of ROS-metabolizing enzymes following activation could contribute to the increase in the ROS concentrations in pDCs.

**Activated pDCs display a high phagosomal pH**. We further aimed to elucidate the mechanisms by which ROS control the capacity of pDCs to cross-prime CD8$^+$ T cells. Knowing that the

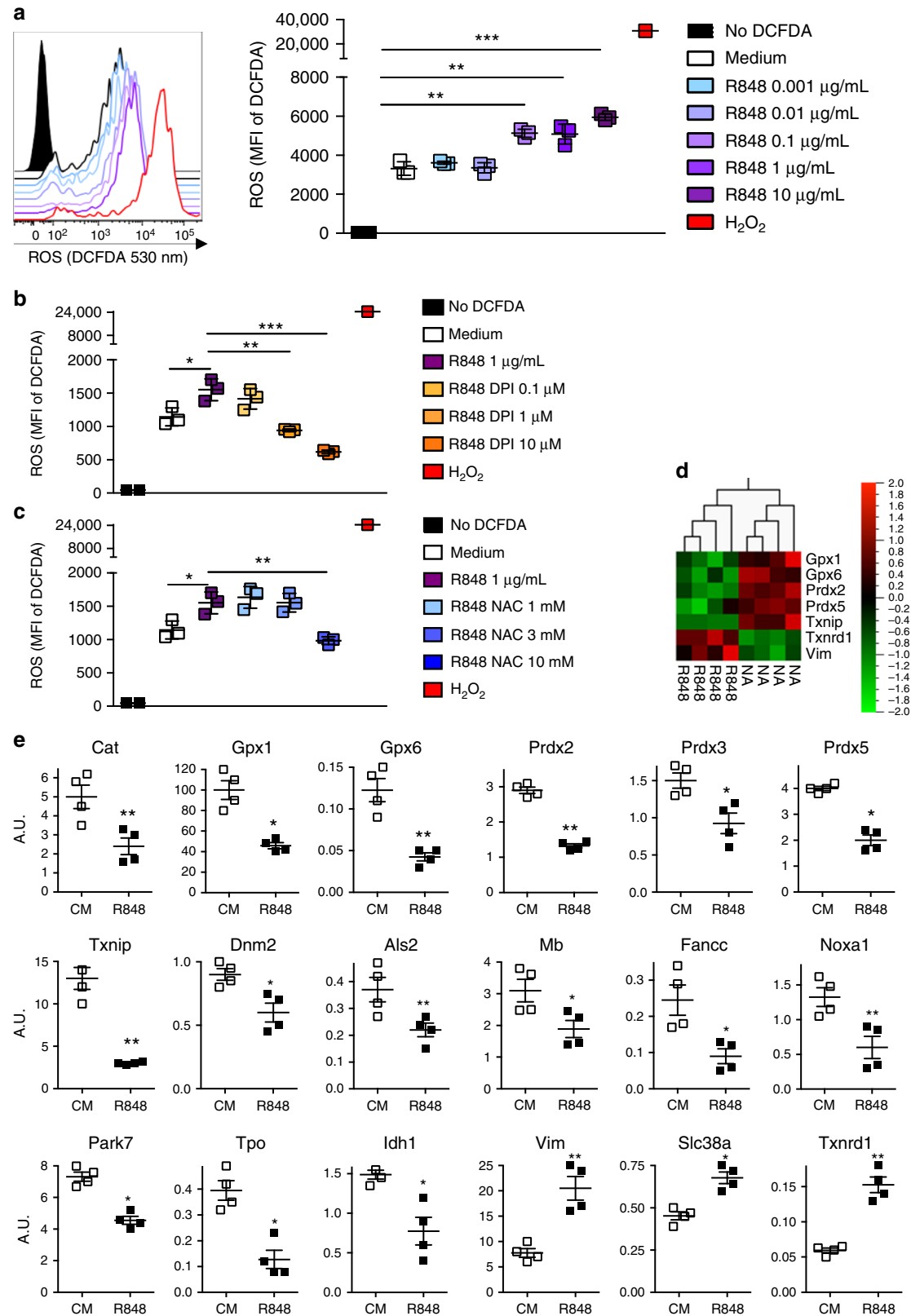

high cross-presenting ability of cDC1s relies on their capacity to maintain a high phagosomal pH that protects the internalized Ags from extensive degradation[9], we, thus, investigated whether the phagosomal pH of the pDCs changes after the activation. Therefore, we performed pH measurements as described by Savina et al.[24] and show in the schematic in Supplementary Fig. 5a. We observed that the phagosomal pH in pDCs cultured in a medium did not change after the bead uptake remaining neutral (Fig. 3a). However, when the pDCs were activated with R848, they alkalinized the phagosome as early as 30 min after the activation, and this alkalinization persisted for at least 2 h after the activation. As previously demonstrated[9], cDC1s alkalinized their phagosomes following the uptake of the beads, whereas CD11b[+] cDCs acidified it (Supplementary Fig. 5b). It is likely that the phagosomal pH alkalinization that was induced by the R848 activation leads to the protection of the internalized Ags. To test this hypothesis, we analyzed the degradation of both the soluble and particulate Ags by resting or R848-activated pDCs. A self-quenched Ag (DQ-OVA) that exhibits a bright fluorescence only upon proteolytic degradation was used as the soluble Ag, whereas OVA-coated beads were used as the particulate Ag. We observed that the R848 treatment induced the protection of both types of Ags from degradation (Fig. 3b).

Previous results from our laboratory have established that the cross-presentation by murine pDCs requires the expression of the transporter TAP[16], suggesting that Ags are processed through the cytosolic pathway. Under this condition, Ags need to be exported from the endosome or the phagosome to the cytosol[4]. Therefore, we assessed whether the R848 stimulation of pDCs could facilitate the transfer of internalized Ags to the cytosol. We used the previously described fluorescence resonance energy transfer (FRET) system to measure the amount of endocytosed β-lactamase that gains access to the cytosol[25,26]. This approach has already been used to measure cytosolic transfer of antigens in pDCs[14]. During the time course allowing 90 min of β-lactamase internalization, no reduction in the enzymatic activity was observed, showing that the β-lactamase is not processed within lysosomes upon endocytosis before translocation into the cytosol[25]. We observed that the R848 activation of pDCs stimulated the β-lactamase export to the cytosol in a significant manner (Fig. 3c). Importantly, the increased Ag access to the cytosol was not due to an increased Ag uptake because, as previously demonstrated with OVA, the R848 activation did not modulate the endocytosis of the soluble β-lactamase. Altogether, these data demonstrate that the induction of cross-presenting capacity of pDCs correlates with a high phagosomal pH, low Ag degradation and increased Ag export to the cytosol as has been described in cDC1s.

**Cross-presentation by pDCs is NOX1- and NOX2-independent.** It is clearly established that the high cross-presenting capacity of

cDC1s is dependent on ROS production by NOX2[9]. To determine the source of ROS in the pDCs, we first analyzed the expression of NOX by RT-qPCR in the steady state and the R848-activated pDCs (Fig. 4a). Among the NOX protein complexes expressed in mice, we detected the expression of NOX1 and 2 in the pDCs, and the R848 activation did not increase their expression, which contrasts with the results obtained for CD69. The R848 activation of pDCs from NOX1[−/−], NOX2[−/−], and NOX1/2[−/−] mice induced an ROS production that was similar to that in C57BL/6 pDCs (Fig. 4b, c). This production was also inhibited by DPI and NAC (Fig. 4d), demonstrating that the R848 activation induced ROS production independently of NOX1 and 2. The Ag protection induced by the R848 activation was also independent of NOX1 and 2 as demonstrated with pDCs from NOX1[−/−], NOX2[−/−], and NOX1/2[−/−] mice (Fig. 4e). Furthermore, the R848-activated NOX1[−/−], NOX2[−/−] (Fig. 4f), and NOX1/2[−/−] (Fig. 4g, h) pDCs efficiently cross-presented the OVA protein to OT-I cells, which was inhibited by DPI and NAC (Fig. 4g). In contrast, the cross-presentation capacity of BMDCs and splenic cDCs was clearly dependent on NOX (Supplementary Fig. 6a, b) as previously described[19]. Altogether, these data demonstrate that, in contrast to BMDCs and cDCs, NOX1/2 are not the source of ROS in pDCs, allowing the activated pDCs to cross-present exogenous Ags to CD8[+] T cells.

**Mitochondrial ROS regulate pDCs cross-presentation capacity.** Mitochondria are an important source of ROS within most mammalian cells[27]. Since the induction of cross-presentation is independent of NADPH oxidases, we then analyzed whether the ROS regulating the cross-presentation capacity of pDCs are produced by the mitochondria. We, thus, analyzed the ROS production by mitochondria (mROS) of pDCs following TLR-L activation using MitoSOX, which is a fluorescent mitochondrial super oxide indicator. At the steady state, we detected a basal production of mROS by pDCs (Fig. 5a). After activation by R848 or CpG, a high increase in mROS was observed (Fig. 5a and Supplementary Fig. 4d), which was strongly inhibited by both DPI and NAC and by the superoxide scavenger S3QEL3 (Fig. 5b). S3QEL3 has been demonstrated to suppress superoxide production by the complex III in the mitochondria without affecting normal electron flux or cellular oxidative phosphorylation[28]. Importantly, S3QEL3 also inhibited the cross-presentation capacity of pDCs induced by R848 (Fig. 5c, d), showing its dependency upon mROS production. This production was independent of NOX1/2 since a significantly higher production of mROS by pDCs from NOX1/2[−/−] mice was observed compared to that in C57BL/6 pDCs (Fig. 5e), which could compensate for the lack of ROS produced by NOX. To confirm that mROS are involved in the induction of cross-presentation, we then used mCAT mice, which overexpress the human catalase targeted to mitochondria, which led to a reduction in mROS[29]. Indeed, mCAT pDCs

**Fig. 2** TLR7 activation induces ROS accumulation in pDCs through the downregulation of ROS-metabolizing enzymes. **a** ROS production in the cytosol of C57BL/6 pDCs was measured by the quantification of the DCFDA fluorescence. The pDCs were loaded with DCFDA and activated with serial concentrations of R848 for 30 min, and then analyzed by flow cytometry. The pDCs incubated for 15 min in $H_2O_2$ and the pDCs that have not been loaded with DCFDA were used as positive and negative controls, respectively. Left panel shows a representative histogram of the DCFDA fluorescence (at 530 nm) on the pDCs treated under these conditions. The cumulative ROS production expressed as the mean MFI ± SD from triplicates is depicted in the right panel and is representative of two independent experiments. **b**, **c** The ROS production by the pDCs activated with R848 (1 μg/mL) was determined as described in **a** in the medium alone or with serial dilutions of DPI (**b**) or NAC (**c**). The results are expressed as the mean MFI ± SD of triplicates and are representative of two experiments. **d** A PCR array was performed on purified pDCs from C57BL/6 mice after 1 h in culture with the medium alone (non-activated, NA) or with R848 (1 μg/mL). The data obtained from four independent experiments were analyzed using the Qlucore software. The results show the 7 heat map obtained with a $p$ value < 0.051 and a $q$ score < 0.104. **e** The PCR array was performed as described in **d**, and the expression of each gene was analyzed individually using an unpaired $t$ test; *$p$ < 0.05; **$p$ < 0.01; ***$p$ < 0.001. The results show the genes that were significantly up- or downregulated upon activation. **a**, **b**, **c**, **e** Each dot represents one replicate

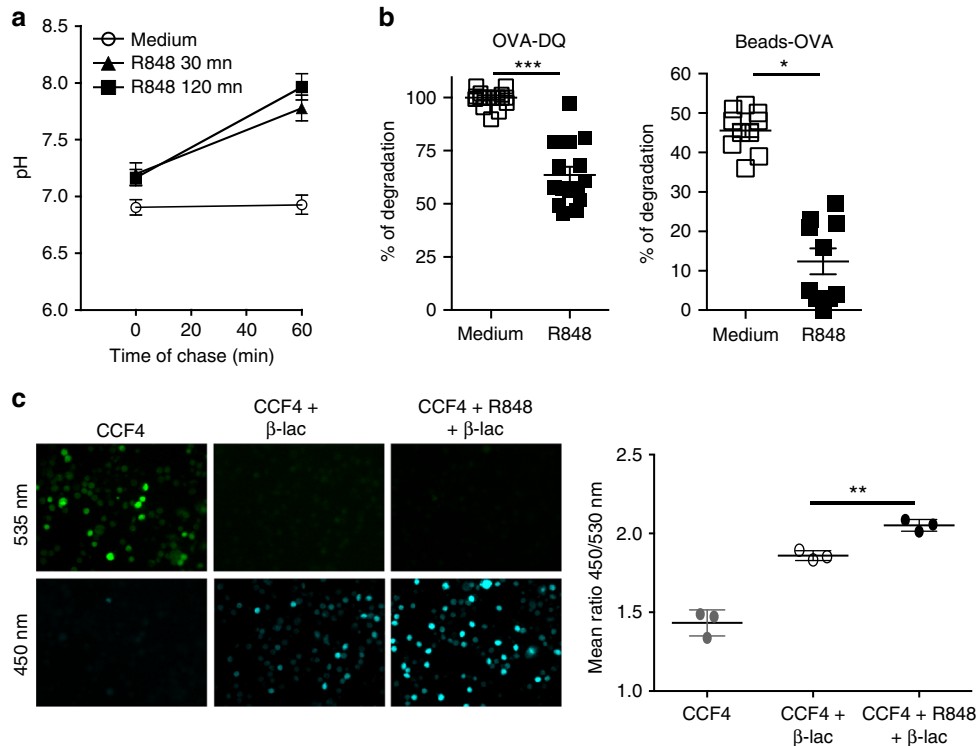

**Fig. 3** Activation of pDCs by the TLR7 ligand leads to a high endosomal pH and Ag protection and transport to the cytosol. **a** Purified C57BL/6 pDCs were either untreated or activated by R848 (1 µg/mL) for 30 or 120 min, and their phagosomal pH was determined. The results are expressed as the mean pH ± SD of triplicates. One representative experiment of 3 is depicted. **b** The degradation of the soluble (OVA-DQ) or particulate (Beads-OVA) Ags by pDCs in the medium alone or with R848 (1 µg/mL) was measured by flow cytometry. The results represent the mean ± SD of triplicates obtained from three independent experiments and each dot represents one replicate. **c** pDCs were purified and cultured on poly-L-lysine for 30 min at room temperature. Then, adherent pDCs were incubated with the β-lactamase substrate CCF4 for 30 min at 37 °C in the medium alone or with R848 (1 µg/mL), and the pDCs were then incubated for 90 min in the presence of β-lactamase (β-lac). The fluorescent intensities were measured at 450 and 535 nm. The images (magnification: ×20) and the corresponding quantification of the fluorescence ratio 450/535 nm are shown and each dot represents one replicate. One representative of three independent experiments is shown. Unpaired $t$ test; $^{*}p < 0.05$; $^{**}p < 0.01$; $^{***}p < 0.001$

produced significantly less mROS compared to C57BL/6 (Fig. 6a), as well as total ROS (Fig. 6b). Although the activation of pDCs from mCAT mice lead to the protection of Ag from degradation as observed for C57BL6 pDCs, the reduction of mROS in pDCs observed in mCAT was associated to higher Ag degradation for both the resting and activated pDCs (Fig. 6c) showing that mROS were involved in the protection of Ag.

We then analyzed the capacity of pDCs from mCAT mice to cross-present OVA after their activation (Fig. 6d, e). The results clearly showed that after activation by R848, the pDCs from mCAT mice were significantly less efficient than C57BL/6 pDCs to cross-present OVA, while the presentation of the SIINFEKL peptide was comparable.

These results clearly indicate that the decrease in mROS production strongly impacted the efficiency of pDCs to cross-present exogenous Ags. In contrast, the capacity of cDCs from mCAT and C57BL/6 mice to cross-present Ags was similar (Supplementary Fig. 6c, d), confirming that cross-presentation by cDCs depends upon the expression of NOX1/2.

We then analyzed the capacity of mCAT mice to develop CD8[+] T-cell responses following in vivo immunization with the OVA soluble antigen with CpG as adjuvant. Indeed, it has been previously established, using Siglec-H deficient mice, that pDCs are required for the in vivo induction of specific CD8[+] T-cell responses by OVA in the presence of CpG[30]. Although no significant difference in the in vivo killing response was observed between mCAT, C57BL/6, and NOX1/2[−/−] mice immunized with OVA/CpG (Fig. 6f), the quantitative analysis of these Ag-specific

CD8[+] T-cell responses by the MHC-I-SIINFEKL dextramers and SIINFEKL-specific IFN-γ ELISPOT showed a strong reduction of the responses of mCAT mice, compared to C57BL/6 and NOX1/2[−/−] mice.

We then analyzed if this strong alteration of T-cell responses of immunized mCAT mice was due to a defect in their immune system. As compared to C57BL/6 and NOX1/2[−/−] mice, mCAT mice showed a similar immune cell distribution (Supplementary Fig. 7a). In addition, the innate responses of these mice were similar to C57BL/6 and NOX1/2[−/−] responses, as determined by the pattern of cytokines and chemokines produced following in vivo stimulation by CpG (Supplementary Fig. 7b). The deficiency of mCAT mice in mROS also did not affected their capacity to control tumor growth, as demonstrated in mice grafted by either B16-OVA or TC-1 tumor cells (Supplementary Fig. 8), confirming that these mice have a functional immune system. Finally, a lower OVA-specific T-cell response, as compared to C57BL/6 and NOX1/2[−/−] mice, was observed in mCAT mice after transfer of OVA-specific CD8[+] OT-I cells and immunization with OVA/CpG (Supplementary Fig. 9a). Thus, these results fully support the conclusion that the strong reduction of the T-cell responses of mCAT mice in response to immunization with OVA/CpG was not due to an intrinsic defect of the mCAT CD8[+] T cells, but to the lower efficiency of the pDCs to cross-present exogenous Ags.

To confirm the major role of pDCs in the induction of CD8[+] T-cell responses by OVA/CpG, C57BL/6, mCAT, and NOX1/2[−/−] mice were depleted of pDCs by treatment with the anti-CD317

Ab. This depletion of pDCs in C57BL/6 and NOX1/2$^{−/−}$ mice immunized with OVA/CpG was associated with a strong reduction of their capacity to develop OVA-specific T-cell responses (Supplementary Fig. 9b), confirming that in this experimental setting, pDCs play a major role in the induction of the OVA-specific CD8$^{+}$ T-cell responses. The low, but detectable, CD8$^{+}$ T-cell response induced in immunized mCAT mice was not affected by the depletion of pDCs, suggesting that these responses were induced by cDCs.

Thus, the reduction of the mROS production in mCAT pDCs greatly impacted the induction of CD8$^{+}$ T-cell responses and

revealed the crucial role of mROS in the induction of CD8$^{+}$ T-cell responses through their capacity to regulate the cross-presentation ability of pDCs.

**mCAT pDCs respond to R848 despite reduced mROS production.** ROS are involved in different functional processes, and several studies have reported a role of ROS in facilitating T-cell responses through the induction of APC maturation[31–33]. Thus, we then assessed the involvement of ROS in pDC activation by analyzing the upregulation of activation markers and

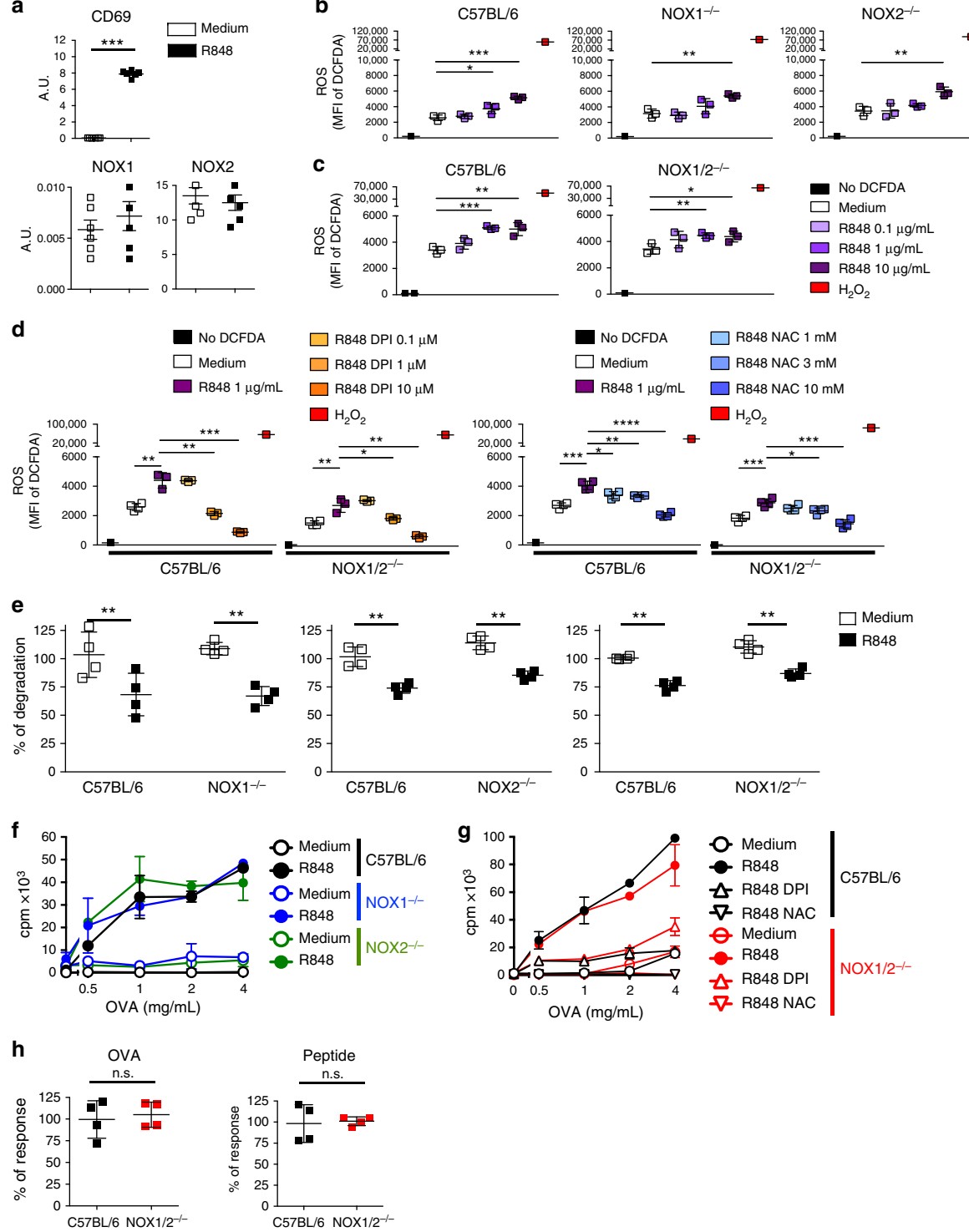

costimulatory molecules, such as CD69, CD40, CD80 and CD86, and MHC-II molecules. As expected, the R848 stimulation induced a strong increase in the expression of MHC-II and costimulatory molecules by pDCs (Fig. 7a). The upregulation of these molecules was strongly affected by DPI and NAC, showing that the pDC activation and maturation are ROS dependent. DPI and NAC were also observed to inhibit the upregulation of these molecules in cDCs following LPS stimulation but with a less marked effect on certain markers, compared to those in the pDCs.

To analyze the role of mROS on the activation and maturation of the DC subsets, we then analyzed the maturation of the pDCs and cDCs from the mCAT mice (Fig. 7b). No significant difference was observed in the upregulation of the activation markers among the pDCs from the mCAT, NOX1/2$^{-/-}$, and C57BL/6 mice after the R848 activation. Similar results were obtained with LPS-activated cDCs.

One of the main functions of pDCs is the production of I-IFNs. We, thus, addressed the role of ROS in the production of IFN-α. DPI and NAC strongly reduced the production of IFN-α by pDCs stimulated with CpG (Fig. 7c). However, no significant difference in the production of IFN-α was observed among the pDCs from the C57BL/6, NOX1/2$^{-/-}$, and mCAT mice (Fig. 7d). Furthermore, we did not detect major differences in the gene expression of R848-activated pDCs from C57BL/6 and mCAT mice (Supplementary Fig. 10), suggesting that the pDCs from these mice respond similarly to this stimulation.

Altogether, these data demonstrate that the reduced mROS production of mCAT mice following TLR7 activation affects only the cross-presentation capacity of pDCs, but does not modify other immune functions.

## Discussion

In the last decade, several studies have established the crucial role of metabolic substrates and the associated pathways in the regulation of immune responses[34] indicating that mitochondrial-dependent signaling can control innate and adaptive immune responses. Consistent with these emerging developments, here, we describe a new regulatory pathway linking mitochondrial metabolism and antigen cross-presentation through the production of ROS in TLR-stimulated pDCs. The reduction of ROS produced by mitochondrial pDCs strongly affects their capacity to trigger CD8$^+$ T-cell responses following immunization, but not their capacity to produce cytokines or chemokines.

The ability of both mouse- and human-activated pDCs to cross-present Ags has been clearly demonstrated[12,14,16,17] and appear to be mainly involved in T-cell activation at the inflammation site participating in body integrity during infection. As previously demonstrated for cDC1s[6], here, we show that pDCs regulate cross-presentation capacity by the alkalization of phagosomal pH and Ag protection that suggests a common mechanism shared by all DC subsets.

cDCs control the phagosomal pH through the consumption of protons by ROS, which is driven by the Rab27a-dependent recruitment of NOX2 to the phagosomes[6]. Accordingly, our results demonstrate that the induction of cross-presentation by pDCs by TLR activation is associated with an increase in ROS production. However, despite the expression of the NADPH oxidases NOX1 and NOX2 by pDCs, neither NOX1 nor NOX2 are involved in the production of ROS following activation.

These observations indicate that another source of ROS is involved in the regulation of cross-presentation by pDCs. Mitochondria metabolism is known to be an important source of H$_2$O$_2$ in most cell types, and it is well established that the activation of immune cells leads to the production of mROS. Indeed, B[35,36] and T[37] cell receptor stimulation leads to mROS production, driving B- and T-cell proliferation, respectively. Importantly, TLR activation also induces the production of mROS in macrophages that participate in the signaling pathway[38].

Here, we show for the first time that the TLR activation in pDCs induces the production of mROS, which is consistent with a recent study that showed that TLR ligation induces an increase in pDC mitochondrial respiration[39]. The production of mROS by the R848-activated pDCs is strongly reduced in the presence of the NAC and S3QEL2 which are ROS and mROS scavengers, respectively, but also by DPI, which has been mainly described as a specific inhibitor of flavoenzymes, such as NOX2, but is also a potent inhibitor of mitochondrial complex I in the respiratory chain[40].

The pDC activation by R848 is associated with the downregulation of several intracellular antioxidant enzyme genes. Some of them, such as the glutathione peroxydases 1 and 6 and the peroxiredoxins 2, 3 and 5 are linked to the mitochondria, that could maintain the level of ROS produced by the mitochondria following the pDC activation over time.

Importantly, for the first time, we associate the production of mROS by pDCs to their activation. To analyze the role played by mROS in the regulation of pDC functions, we used mCAT transgenic mice that express the human catalase in the mitochondria[29], leading to significantly lower mitochondrial H$_2$O$_2$ levels[29,38]. Consistently with previous findings, a lower amount of mROS is produced in the mCAT-activated pDCs compared to C57BL/6 pDCs. However, these mice do not present any defect in the composition of their immune cells or of their innate responses to TLR-L activation. In contrast, we observe a strong reduction in the in vitro cross-presentation capacity of mCAT pDCs following TLR activation and a marked reduction of their CD8$^+$ T-cell responses induced following in vivo immunization with OVA/CpG. These results demonstrate that the ROS produced by mitochondria regulate the cross-presentation ability of pDCs and

**Fig. 4** NOX1 and NOX2 are not required for Ag protection and cross-presentation by pDCs. **a** Purified pDCs from C57BL/6 mice were cultured for 1 h in the medium alone or with R848 (1 μg/mL), and the *cd69, nox1*, and *nox2* gene expression was assessed by RT-PCR. Cumulative data from three independent experiments are shown as the mean ± SD of duplicates. **b, c** The ROS production in the cytosol of the pDCs purified from C57BL/6, NOX1$^{-/-}$, and NOX2$^{-/-}$ (**b**) or C57BL/6 and NOX1/2$^{-/-}$ (**c**) mice was measured by the quantification of the DCFDA fluorescence. The ROS production is expressed as the mean MFI ± SD from triplicates and is representative of two independent experiments. **d** The ROS production by pDCs from the C57BL/6 and NOX1/2$^{-/-}$ mice activated with R848 (1 μg/mL) in the medium alone or with serial dilutions of DPI or NAC. The results are expressed as the mean MFI ± SD of triplicates and are representative of two experiments. **e** The OVA-DQ degradation by the pDCs purified from the C57BL/6, NOX1$^{-/-}$, NOX2$^{-/-}$, and NOX1/2$^{-/-}$ mice activated or not for 30 min by R848 (1 μg/mL) was analyzed by flow cytometry. The data represent the mean ± SD of triplicates from four independent experiments and each dot represents one replicate. **f–h** The pDCs were purified from C57BL/6, NOX1$^{-/-}$ or NOX2$^{-/-}$ mice (**f**) or C57BL/6 and NOX1/2$^{-/-}$ mice (**g**) and the OVA-cross-presentation to LN OT-I cells was assessed in the medium alone or with R848 (1 μg/mL), in the presence of DPI (0.1 μM) or NAC (10 mM) (**g**). The results are expressed as the mean cpm ± SD of triplicates. One representative experiment of 3 is depicted. **h** The results show the cumulative data from four experiments using R848-activated pDCs from C57BL/6 and NOX1/2$^{-/-}$ mice loaded with the OVA protein (4 mg/mL) or the SIINFEKL OVA peptide (1 μg/mL) and are expressed as the mean ± SD percentage of the response obtained with C57BL/6 pDCs. Each dot represents the mean of duplicates from one experiment. Unpaired *t* test; *$p < 0.05$; **$p < 0.01$; ***$p < 0.001$; ****$p<0.0001$

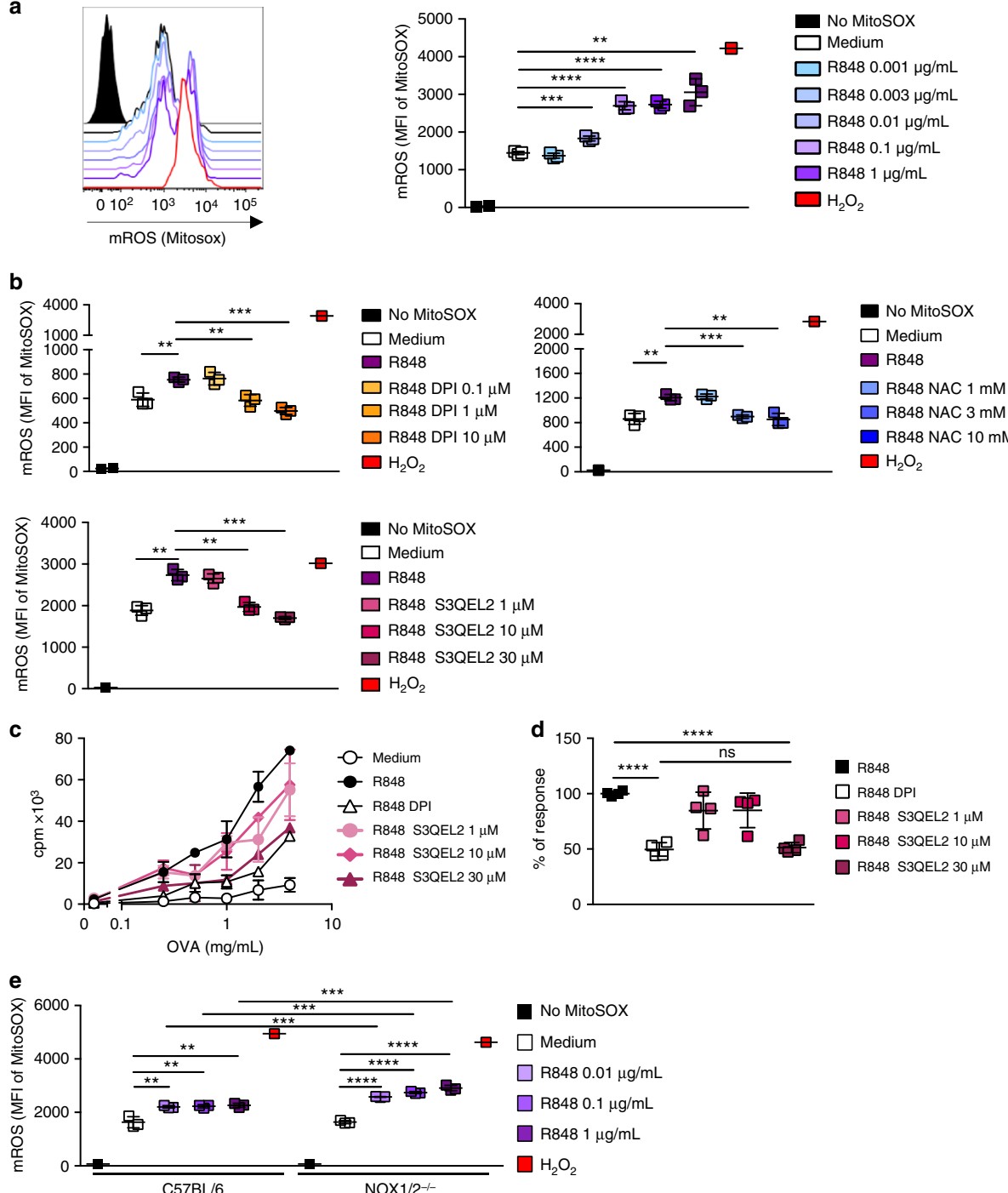

**Fig. 5** Ag cross-presentation by pDCs is dependent on mitochondrial ROS production. **a**, **b** Production of mROS by pDCs was measured by quantification of the MitoSOX fluorescence. **a** pDCs purified from C57BL/6 mice were activated with R848 for 30 min and analyzed by flow cytometry. The left panel shows a representative histogram of the MitoSOX fluorescence on resting or R848-activated pDCs. The right panel shows cumulative results from triplicates and is representative of two independent experiments. **b** The production of mROS by pDCs activated with R848 (1 μg/mL) was determined as described in **a** in medium alone or with serial dilutions of DPI, NAC or S3QEL2. The results are expressed as the mean MFI ± SD of triplicates and are representative of two experiments. **c**, **d** pDCs were purified from C57BL/6 mice and the cross-presentation of OVA to LN OT-I cells was assessed in medium alone or with R848 (1 μg/mL) in the presence of DPI (0.1 μM) or serial dilutions of S3QEL2. **c** The results are expressed as the mean cpm ± SD of triplicates. One representative experiment of 2 is depicted. **d** The results show the cumulative data from two experiments using R848-activated pDCs, loaded with the OVA protein (4 mg/mL) and are expressed as the mean ± SD percentage of the response obtained with R848-activated pDCs in the absence of inhibitor. **e** mROS production by C57BL/6 and NOX1/2$^{-/-}$ pDCs activated for 30 min with R848. **a**, **b**, **d**, **e** Each dot represents one replicate. Unpaired $t$ test; n.s., non-significant, *$p < 0.05$; **$p < 0.01$; ***$p < 0.001$; ****$p < 0.0001$

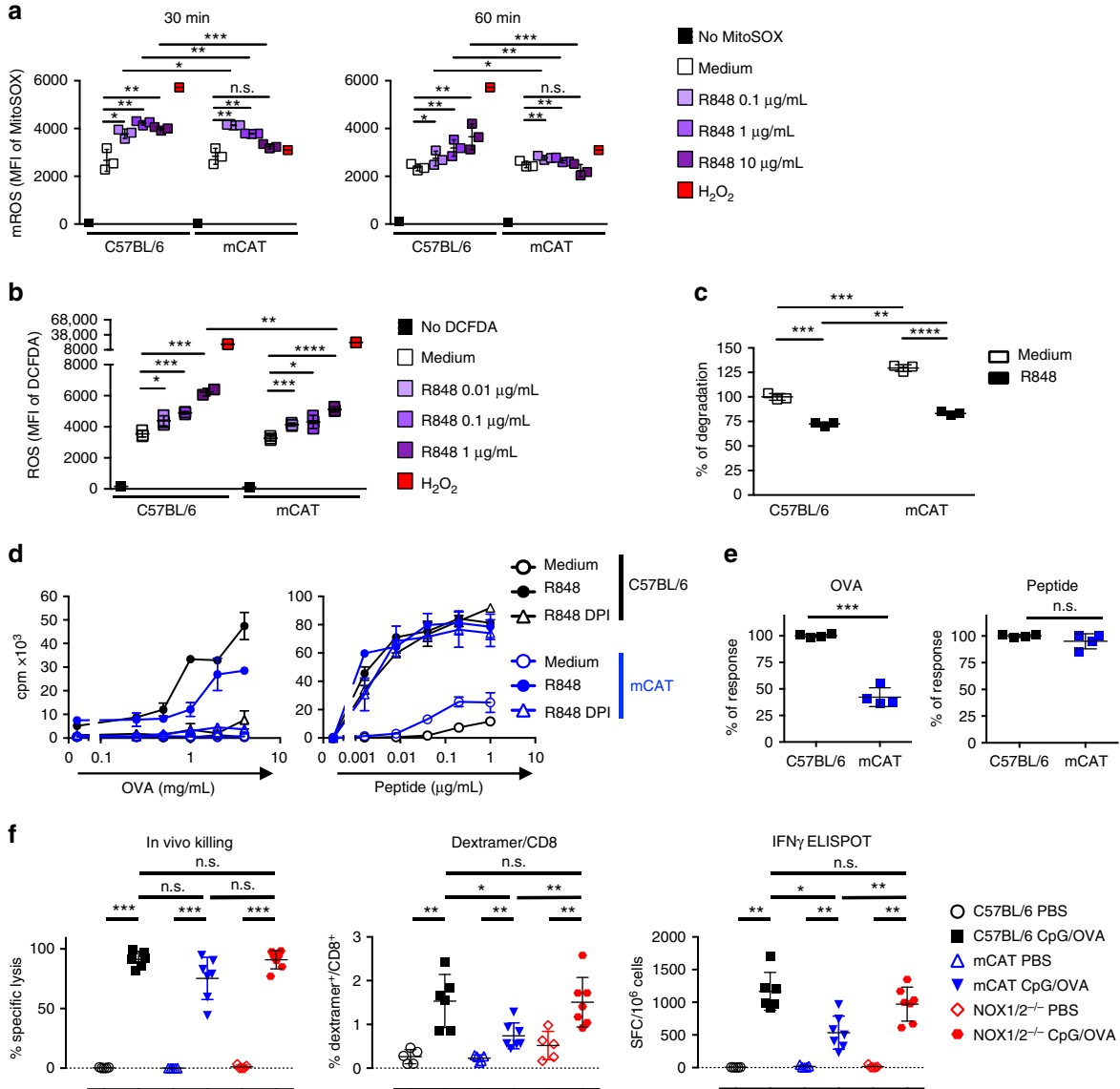

**Fig. 6** The reduced production of mROS by pDCs from mCAT mice is associated with poor Ag cross-presentation efficacy. **a**, **b** Production of mROS by C57BL/6 and mCAT pDCs activated for 30 (**a**, **b**) or 60 min (**a**) with R848. Results are expressed as the mean MFI ± SD of triplicates and are representative of two experiments. **c** The degradation of OVA-DQ by C57BL/6 or mCAT pDCs activated for 30 min by R848 (1 μg/mL) was analyzed by flow cytometry. The data represent the mean ± SD of triplicates from three independent experiments. **a**–**c** Each dot represents one replicate. **d**, **e** The cross-presentation of OVA by C57BL/6 or mCAT pDCs was assessed in medium or with R848 in the presence or the absence of DPI (0.1 μM). **d** The results are shown as the mean cpm ± SD of triplicates and are representative of three experiments. **e** The results show the cumulative data from four experiments using R848-activated pDCs from C57BL/6 and mCAT mice, loaded with the OVA protein (4 mg/mL) or the SIINFEKL OVA peptide (1 μg/mL) in the absence of DPI and are expressed as the mean ± SD percentage of the response obtained with C57BL/6 pDCs. Each dot represents the mean of duplicates of one experiment. **f** C57BL/6, mCAT, and NOX1/2$^{-/-}$ mice were immunized i.v. with 100 μg of OVA mixed with 30 μg CpG /DOTAP. Seven days later, the anti-OVA CD8$^+$ T-cell response was assessed by an in vivo killing assay, SIINFEKL/H2-K$^b$ dextramer staining, and IFN-γ ELISPOT. The results are expressed as the percentage of OVA-specific lysis for CTL activity, the percentage of SIINFEKL/H2-K$^b$ dextramer$^+$ among total CD8$^+$ splenocytes for dextramer staining and IFN-γ spot-forming cells (SFC) per 10$^6$ splenocytes for ELISPOT. Each dot represents individual mice. Unpaired t test; n.s., non-significant, *$p < 0.05$; **$p < 0.01$; ***$p < 0.001$; ****$p < 0.0001$

play a crucial role in their capacity to induce CD8$^+$ T-cell response against exogenous Ags in vivo.

Indeed, we show that the production of mROS is associated with the regulation of the phagosomal pH, leading to Ag protection and export to the cytosol. Importantly, the reduction of mROS in mCAT mice leads to a high degration of Ag in pDCs associated to low cross-presentation ability. To regulate the phagosomal pH of the activated pDCs, the ROS produced by the mitochondria should be transferred to the phagosomes. In macrophages, the engagement of TLR/MyD88 was shown to

result in the recruitment of mitochondria to the phagosome, which was associated with an increase in mROS production[38]. mROS are then transferred to the phagosomes, where they directly contribute to microbial killing. This process involves the translocation of the TRAF6 adaptor to the mitochondria, where it engages the protein ECSIT (Evolutionary Conserved Signaling Intermediate in Toll pathways), which is implicated in the respiratory chain assembly. The activation of the pDCs following the TLR ligation involves both the MyD88 and TRAF6 adaptors, suggesting that the transfer of mROS to the phagosome follows

the same process. This hypothesis is supported by our demonstration of the upregulation of vimentin (Vim) and thioredoxin reductase 1 (Txnrd1) in the R848-activated pDCs, which are, respectively, involved in the regulation of mitochondria motility[41] and the induction of actin and tubulin polymerization[42]. These two molecules modulate organelle movements in the cytosol and, thus, could be involved in the recruitment of mitochondria to the phagosome.

We also observe that ROS are involved in cDC maturation, which is consistent with previous results that showed that the endogenous ROS production by Kupffer cells regulates the expression of CD40, CD80, and MHC class II molecules[33] and

that an antioxidant treatment abolishes the LPS-induced expression of costimulatory molecules by BMDCs[32].

However, recent studies demonstrate that after activation, pDCs use mitochondrial respiration, through an autocrine I-IFN receptor-dependent pathway while cDCs increase glycolysis[39,43]. Interestingly, in this process, I-IFN promotes oxidative phosphorylation, which is required for full pDC activation. These data, including our work, suggest that cDCs and pDCs may use different ROS pathways following activation.

Altogether, our results establish that pDCs regulate cross-presentation through the regulation of their phagosomal pH. In contrast to cDCs, the ROS involved in this process are generated

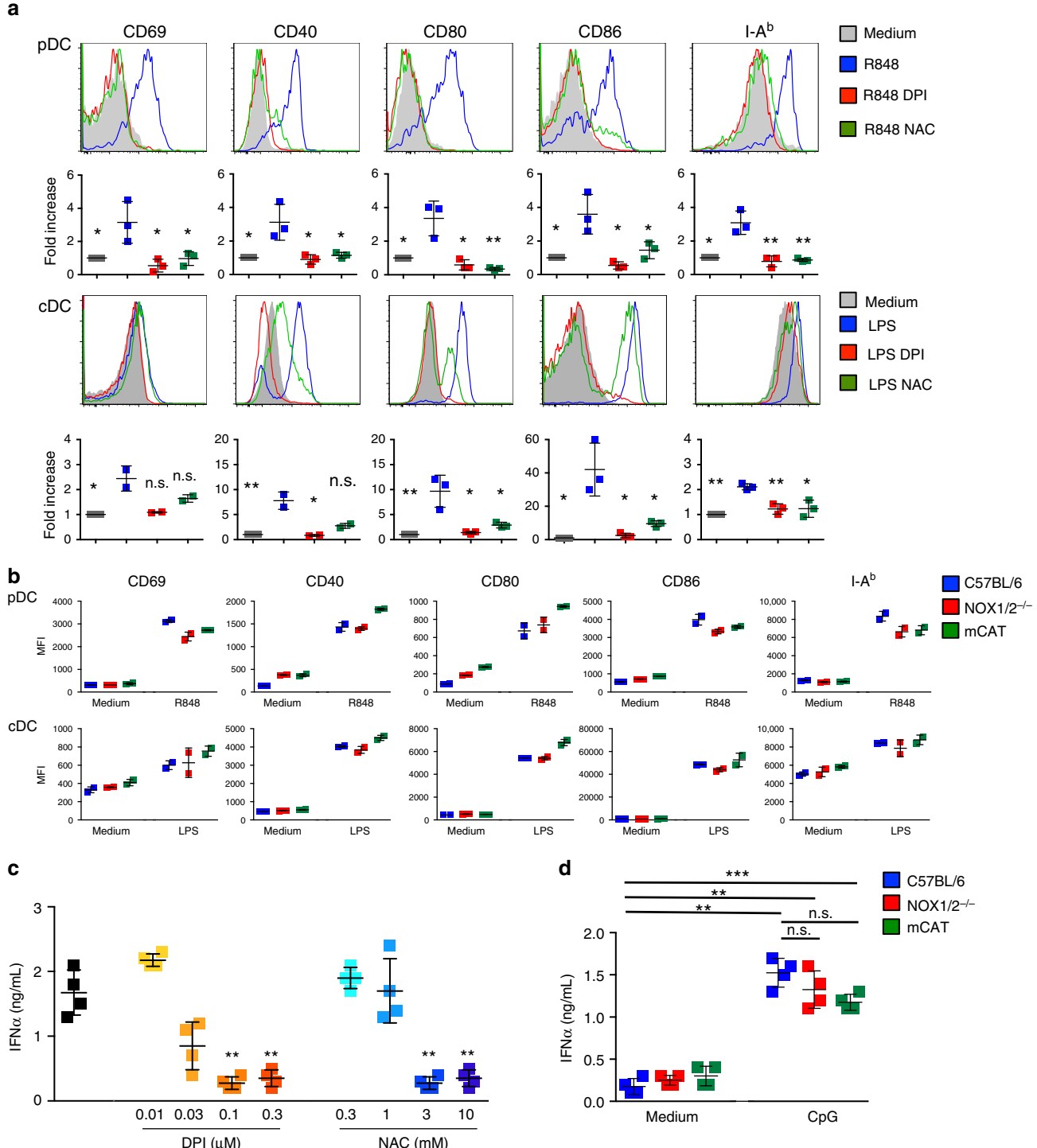

by the mitochondrial respiratory chain, confirming that pDCs use different metabolic pathways from cDCs to support their specialized functions[43].

The present study also demonstrates that the role of ROS in pDCs is not restricted to Ag cross-presentation. Indeed, both the I-IFN production and the upregulation of the maturation markers induced following pDC activation are abolished by DPI and NAC. However, the maturation of pDCs is not affected in mCAT or NOX1/2$^{-/-}$ mice, suggesting a redundancy in the source of ROS.

Until now, the studies analyzing the role of pDCs have demonstrated that pDCs are mainly involved in the induction of immune responses through their capacity to produce cytokines and chemokines, and particularly I-IFNs[30,44–46]. However, the present study demonstrates that the strong decrease of CD8$^+$ T-cell responses observed in mCAT mice following immunization with the OVA soluble antigen is due to a selective defect in their pDCs cross-presentation efficiency, rather than to a general defect of their innate responses and of their capacity to produce cytokines and chemokines.

To the best of our knowledge, this work describes for the first time the involvement of the mitochondria in Ag presentation and highlights the importance of mitochondrial metabolism in pDC biology, particularly in the induction of CD8$^+$ T-cell responses following immunization, opening the way for the development of new immunotherapeutic strategies that target pDCs.

## Methods

**Mice.** The 129/sv (H-2$^b$), C57BL/6 (H-2$^b$), and BALB/c (H-2$^d$) mice were purchased from Charles River (L'Arbesle, France). The Rag$^{-/-}$ OT-I T-cell receptor transgenic mice, specific for the K$^b$-restricted OVA$_{257-264}$ (SIINFEKL) epitope, were provided by the animal facilities of the Pasteur Institute. The Act-mOVA/Kb$^{-/-}$ transgenic mice on a C57BL/6 background, expressing membrane-bound OVA and lacking MHC class I (mOVA Kb$^{-/-}$), were a kind gift from Matthew Albert (Pasteur Institute). The B6.129X1-Nox1$^{m1Kkr}$/J C57BL/6 mice, lacking the NOX1 NADPH (NOX1$^{-/-}$), or the B6.129-Cybb$^{tm1Din}$/J mice, lacking the NOX2 subunit gp91 (NOX2$^{-/-}$) from Jackson Laboratory, were purchased from Charles River. The double NOX1/2 KO mice (NOX1/2$^{-/-}$) were obtained by backcrossing the NOX1$^{-/-}$ and NOX2$^{-/-}$ mice. The B6.Cg-Tg(CAG-OTC/CAT) 4033Prab/J transgenic mice from Jackson Laboratory, expressing the human catalase gene in the mitochondria (mCAT), were purchased from Charles River. All mice were bred at the animal facilities of the Pasteur Institute, maintained under specific pathogen-free conditions and used between 6 and 12 weeks old. Studies using mice were validated by the CETEA ethics committee number 89 (Institut Pasteur, Paris, France) and by the French Ministry of Research (number HA0001).

**Cell culture and reagents.** RPMI 1640, containing Glutamax, and HBSS were obtained from Life Technologies (Fisher, Paisley, United Kingdom). The complete medium (CM) consisted of RPMI supplemented with 10% fetal calf serum (Valeant Pharmaceuticals, Costa Mesa, CA), $5\times10^{-5}$ M 2-mercaptoethanol, and antibiotics (penicillin 100 U/mL, streptomycin 100 μg/mL, Life technologies). The TLR-agonists R848 (InvivoGen, Toulouse, France) and CpG 1826 (Sigma-Aldrich, Saint Quentin Fallavier, France) were used at 1 μg/mL and 10 μg/mL, respectively. The OVA protein was obtained from Sigma-Aldrich. The synthetic OVA peptide SIINFEKL corresponding to the H-2K$^b$-restricted CTL epitope of OVA was

purchased from NeoMPS (Strasbourg, France). Diphenyleneiodonium (DPI), plumbagin, apocynin, and N-acetyl-L-cysteine (NAC) were purchased from Sigma-Aldrich. S3QEL2 was purchased from Bio-Techne (Lille, France).

**Cell purification.** Purified pDCs were obtained either by magnetic selection (AutoMACS Pro, Miltenyi Biotec) or FACS Aria III (BioLife Sciences, Le Pont de Claix, France) sorting. For the magnetic pDC purification, splenocytes were depleted from CD11b$^+$ and CD19$^+$ cells using MACS anti-CD11b and anti-CD19 microbeads before the positive selection of PDCA1$^+$ cells by AutoMACS. For the FACS-sorted pDCs, magnetic purified pDCs were stained with anti-CD11c, anti-B220, and anti-PDCA1 antibodies, and the CD11c$^{low}$B220$^+$PDCA1$^+$ fraction was sorted using a FACS Aria III (Supplementary Fig. 11). The purified cDCs were obtained after the MACS anti-CD11c labeling and positive selection by AutoMACS.

**Measure of ROS production.** The total ROS production was assessed using the splenic pDCs purified by PDCA1 magnetic positive selection by AutoMACS. The pDCs were incubated at 37 °C in RPMI supplemented with 10% fetal calf serum, with or without DPI for the indicated concentration. After washing with HBSS, the pDCs were loaded with 5 μM DCFDA (Sigma-Aldrich) in HBSS for 30 min at 37 °C, washed and then cultured in complete medium at 37 °C with or without R848 or NAC for the indicated concentration. After staining with the anti-PDCA-1 and -CD11c antibodies, ROS production by the pDCs was determined by measuring the fluorescence at 530 nm on PDCA1$^+$ CD11c$^{low}$ live cells.

Mitochondrial ROS (mROS) was measured using the magnetically purified splenic pDCs cultured for 30 min in the complete medium at 37 °C with or without R848 and DPI or NAC at the indicated concentration and time. After washing with HBBS, the cells were loaded with 5 μM MitoSOX (Molecular Probes, Illkirch, France) in HBSS for 30 min at 37 °C. After staining with the anti-PDCA-1 and -CD11c antibodies, mROS production by pDCs was determined by measuring the fluorescence at 580 nm on PDCA1$^+$ CD11c$^{low}$ live cells.

**RT-PCR assay.** Magnetically purified splenic pDCs were incubated for 1 h at 37 °C in the presence or absence of R848. Then, the pDCs were FACS sorted and lysed, and the RNA extraction was performed using the RNeasy Plus Micro Kit (Qiagen). For the reverse transcription, SuperScript II RT from GIBCO-BRL (San Francisco, USA) was used. QuantiTect primers for GAPDH (Mm_Gapdh_3_SG), NOX1 (Mm_Nox1_1_SG), NOX2 (Mm_Nox2_1_SG), and CD69 (Mm_Cd69_1_SG) were purchased from Qiagen, and the assays were performed according to the manufacturer's instructions using the SsoFast EVA green SuperMix (Bio-Rad, Marnes-la-Coquette, France). The mouse oxidative stress RT$^2$ Profiler PCR array from Qiagen (SA-Bioscience) was performed as indicated by the provider, and the gene expression in resting and R848-activated pDCs was normalized to an average of five housekeeping genes (Gusb, Hprt, Hsp90ab1, Gapdh, and Actb). The data were analyzed using the Qlucore software (Lund, Sweden). All RT-PCR experiments were run on CFX96-Real Time System (Bio-Rad).

**Ag presentation assay.** The particulate Ag corresponded to 1 μm beads (polystyrene microspheres) coated with the OVA protein at 0.5 mg/mL by passive adsorption according to the manufacturer's procedure (Polysciences, Warrington, PA, USA). The exogenous cellular Ag originated from the H-2$^d$ splenocytes from the BALB/c mice that were pulsed with OVA (10 mg/mL) for 2 h or mOVA Kb$^{-/-}$ splenocytes. The splenocytes were incubated in the presence of camptothecin (1 μM) for 2 h to induce apoptosis. Magnetically purified splenic pDCs ($1–2.5\times10^4$/well) were first incubated for 1 h with serial dilutions of Ags in the presence or absence of 1 μg/mL R848 and the inhibitors and $5\times10^4$ OT-I cells/well were added after washing. Then, 72 h later, the T-cell proliferation was scored according to [$^3$H]-thymidine incorporation as previously described[47].

**Fig. 7** ROS control the activation of pDCs and their capacity to produce IFN-I. **a** Purified pDCs (upper panels) or cDCs (lower panels) from C57BL/6 mice were incubated at 37 °C in the medium alone or with R848 (1 μg/mL) or LPS (1 μg/mL) in the presence of either DPI (0.1 μM) or NAC (10 mM). After 60 min, the cells were washed and incubated in the medium alone for 18 h. The expression of the maturation markers CD69, CD40, CD80, CD86 and I-A$^b$ on CD11b$^-$PDCA1$^+$CD11c$^{low}$ pDCs or CD11c$^{high}$ cDCs was determined by flow cytometry. The upper panels show histograms from one representative experiment, and the lower panels show the cumulative results from three independent experiments. The results are expressed as the mean ± SD fold increase of the fluorescence intensity compared to that in the unstimulated pDCs or cDCs. **b** Purified pDCs (upper panels) or cDCs (lower panels) from C57BL/6, NOX1/2$^{-/-}$ or mCAT mice were incubated at 37 °C in the medium alone or with either R848 (1 μg/mL) or LPS (1 μg/mL). After 60 min, the cells were washed and incubated in the medium alone for 18 h. The expression of the maturation markers CD69, CD40, CD80, CD86 and I-A$^b$ on CD11b$^-$PDCA1$^+$CD11c$^{low}$ pDCs or CD11c$^{high}$ cDCs was determined by flow cytometry. The results are expressed as the mean ± SD MFI and are representative of two independent experiments. **c** Purified pDCs from C57BL/6 mice were incubated at 37 °C for 1 h with CpG (10 μg/mL) in the absence or the presence of serial dilutions of DPI or NAC and then washed and incubated for 23 h. Then, the supernatants were collected, and IFN-α was analyzed by ELISA. One representative experiment of 2 is depicted. **d** Purified pDCs from C57BL/6, NOX1/2$^{-/-}$, or mCAT mice were incubated for 1 h in the medium alone or with CpG (10 μg/mL) and then washed and incubated for 23 h. Then, the supernatants were collected, and IFN-α was dosed by ELISA. One representative experiment of 2 is depicted. Each dot represents one replicate. Unpaired $t$ test; n.s., non-significant, *$p < 0.05$; **$p < 0.01$; ***$p < 0.001$

**Type I IFN production**. The purified pDCs ($10^5$/well) were incubated at 37 °C for 1 h with or without CpG 2216 and DPI, and then, the cells were washed and incubated in complete medium at 37 °C. After 23 h, the supernatants were collected, and IFN-α was dosed by an ELISA using capture (clone RMMA-1) and detection (polyclonal anti-IFN-α) antibodies from R&D Systems (Lille, France). The revelation was performed using the HRP-IgG Donkey anti-rabbit antibody (Southern Biotec).

**Quantification of cytokines and chemokines**. Cytokines and chemokines were quantified in mouse serum using the 36-plex immunoassays (Luminex, Affymetrix) following the manufacturer's procedure. The data were acquired on a validated and calibrated Bio-Plex 200 system (Bio-Rad) and were analyzed using Bio-Plex Manager 6.0 software (Bio-Rad).

**Ag export to the cytosol by FRET**. The protocol has been described in detail in Keller et al.[48]. Briefly, the purified pDCs were loaded with 6 μM of the β-lactamase substrate CCF4 (AM LiveBLAzer, Invitrogen) in an EM LiveBLAzer buffer (120 mM NaCl, 7 mM KCl, 1.8 mM CaCl$_2$, 0.8 mM MgCl$_2$, 5 mM Glucose, and 25 mM Hepes at pH 7.3) and 4% of the LiveBLAzer solution B (Invitrogen). Then, the pDCs were incubated for 90 min with 1 mg/mL of β-lactamase in the supplemented EM LiveBLAzer buffer. The cells were observed under a wide-field epifluorescence microscope (Nikon TI) at ×20 magnification. The 405 nm wavelength was used for the excitation of the CCF4 compound. Emissions at 535 nm (FRET, green/intact probe) and 450 nm (no FRET, blue/cleaved probe) were analyzed in the absence or presence of β-lactamase with or without the R848 activation. The ratiometric values (450:535) were calculated using a dedicated macro for the image analysis software Metamorph.

**Ag degradation assays**. Magnetically purified splenic pDCs were incubated for 30 min at 37 °C in the complete medium with or without R848 together with the different inhibitors. For the soluble Ag degradation assays, half of the cells was incubated at 37 °C with 0.5 mg/mL OVA-DQ (Molecular Probes) together with the different inhibitors for 30 min, whereas the other half was kept at 4 °C in the complete medium containing 0.1 % azide, DQ-OVA and the inhibitors. The cells were washed and stained with antibodies, and the OVA-DQ fluorescence on CD11b⁻PDCA1⁺CD11c^low cells was determined by flow cytometry. The percentage of degradation ($D$) was calculated as follow: $D = $ [(MFI sample 37 °C / MFI sample azide 4 °C) / (MFI control CM 37 °C / MFI control medium azide 4 °C)] × 100. The particulate Ag degradation was determined as previously described[24]. Briefly, following 40 min of the R848 activation in the presence or absence of the inhibitors, the pDCs were incubated with the OVA-coated beads. Then, the cells were lysed, and the remaining OVA on the phagocytosed beads were determined by flow cytometry.

**pH measurement**. The phagosomal pH was determined as previously described[24]. Briefly, 3 μm amino polybeads were covalently coupled with equal amounts of FITC (pH sensitive) and A647 (pH insensitive) in the presence of sodium hydrogen carbonate buffer (pH 8) for 2 h at room temperature. After extensively washing with glycine (100 mM), the beads were suspended in PBS. The pDCs were magnetically purified, activated with R848 for the indicated time and pulsed with the FITC-A647-Beads for 20 min at 37 °C in the complete medium. After washing with a large volume of cold PBS-0.5 % BSA, the cells were suspended in complete medium and either kept on ice ("Pulsed") or incubated at 37 °C for 1 h ("Chased"). The cells were labeled and analyzed by flow cytometry. Gating-on SSC^hiA647⁺CD11c^lowPDCA1⁺ cells allowed for the selection of pDCs that have phagocytosed one bead. The ratio of the MFI emission between the two dyes (A647/FITC) was determined. The values were compared with a standard curve that was obtained by suspending the cells that had phagocytosed beads for 30 min at a fixed pH (ranging from pH 5.5 to 8) in 0.1 % Triton X-100.

**Flow cytometry**. The following antibodies coupled to different fluorochromes were used for the flow cytometry: anti-CD11c (N418), -CD11b (M1/70), -CD8α (53-6.7), -BST-2 (PDCA-1 eBio129c), -CD80 (16-10A1), -CD86 (GL1), and -CD69 (H1.2F3) from eBiosciences (San Diego, CA, USA); -CD19 (ID3), -H2-K^b (AF6-88-5), -I-A^b (AF6-120.1), and -PDC-TREM (4A6) from Biolegend; and -CD40 (HM40-3) from BD Pharmingen. SIINFEKEL/H-2K^b/PE MHC dextramers (Immudex, Copenhagen, Denmark) were used to identify the OVA-specific CD8⁺ T cells. The control isotypes were purchased from the corresponding providers. Cells were acquired on a Fortessa analyzer (BD). The data were analyzed using the FlowJo Software (Tree Star Inc. Stanford, USA).

To visualize PKH67⁺ apoptotic cells in pDCs, cells were stained with Abs and digital imaging was performed on an imaging flow cytometer (ImageStreamX; Merck). At least, $10^4$ pDCs were imaged for each sample and analyzed using manufacturer's software (IDEAS).

**Transfer of OT-I CD8⁺ T cells**. OT-I CD8⁺ T cells were purified by magnetic negative selection using the CD8⁺ T-cell isolation kit from Miltenyi Biotec following the manufacturer's procedure. Then, the cells were labeled 15 min at room temperature with 10 μM CFSE (Molecular Probes) and $5×10^5$ cells were injected to mice by i.v. route. Mice were immunized 1 day later and the CD8⁺ T-cell response was analyzed 4 days after immunization.

**Depletion of pDCs in vivo**. Mice were injected i.p. with 500 μg of the anti-CD317 mAb (InvivoMab clone BX444, BioXcell) or of control isotype 1 day before immunization and then with 250 μg of anti-CD317 mAb or of control isotype on days 1 and 4 after immunization.

**In vivo killing assay**. Naive syngeneic splenocytes were pulsed with the OVA$_{257-264}$ peptide (10 μg/mL) (30 min, 37 °C), washed extensively and labeled with a high concentration (1.25 μM) of CFSE. The non-pulsed control population was labeled with a low concentration (0.125 μM) of CFSE. CFSE^high- and CFSE^low-labeled cells were mixed in a 1:1 ratio ($5×10^6$ cells of each population) and injected intravenously into the mice 6 days after the immunization. The number of CFSE-positive cells remaining in the spleen after 20 h was determined by flow cytometry, and the percentage of specific lysis was calculated as follows:

% specific lysis = 100 – [100 × (%CFSE^high immunized mice / %CFSE^low immunized mice) / (%CFSE^high naive mouse / %CFSE^low naive mouse)].

**ELISPOT assay**. The frequency of the OVA$_{257-264}$-specific IFNγ-producing cells was determined by ELISPOT as described previously[47]. The results are expressed as the number of spot-forming cells (SFC) per million splenocytes. For each mouse, the number of OVA$_{257-264}$-specific IFNγ SFCs was determined by calculating the difference between the number of spots generated in the presence and absence of the OVA$_{257-264}$ peptide (10 μg/mL).

**Transcriptomic analysis of pDCs**. Following purification by FACS Aria III, splenic pDCs were either immediately lysed after purification or following in vitro culture in the absence or the presence of R848. RNAs were extracted using the NucleoSpin® RNA XS kit (Macherey-Nagel) and were quantified using the Qubit fluorometer (Life Technologies). Transcriptomic analysis was performed using the nCounter® PanCancer Immune Profiling Panel (NanoString Technologies) which allowed for the analysis of approximately 800 genes following the manufacturer's procedure. The samples were processed with the fully automated nCounter Prep Station (NanoString Technologies) and were imaged on the nCounter Digital Analyzer (NanoString Technologies). All the samples were normalized using manufacturer's software, with the expression level of Rpl19, Oa71, Eef1g, Ppia, and Polr2a housekeeping genes. The normalized data were then visualized and analyzed with the R software.

**Statistical analyses**. Unless otherwise specified, the statistical analyses were performed using prism software (GraphPad) and an unpaired $t$ test; $*p < 0.05$; $**p < 0.01$; $***p < 0.001$, $****p<0.0001$.

**Data availability**. The data that support the findings of this study are available from the corresponding author upon reasonable request.

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

## Acknowledgements

We thank Matthew Albert and Krause Karl-Heinz for providing the Act-mOVA/Kb$^{-/-}$ transgenic mice and the NOX1$^{-/-}$ mice respectively. This work was supported by the Ligue Nationale Contre le Cancer (Equipe Labellisée 2017). J.E. acknowledges funding by the ERC (Consolidator grant EndoSubvert) and the ANR (program StopBugEntry and AutoHostPath).

## Author contributions

M.O. designed and performed the experiments, analyzed the results and provided valuable inputs regarding the manuscript. C.G. and J.M. designed and performed the experiments, analyzed the results and wrote the manuscript. P.R. and C.F. performed experiments and analyzed the results. A.B. designed and performed the experiments under the supervision of J.E. A.S. and S.A. discussed the data. E.O.-D. provided the NOX1$^{-/-}$ mice. C.L. and G.D. conceived and supervised the study, designed the experiments, analyzed and discussed the data, and wrote the manuscript. G.D. also designed and performed the experiments.

## Additional information

**Competing interests:** The authors declare no competing interests.

