## [Peer Review File · Nature Communications]

Reviewers' comments:

Reviewer #1 (ROS, dendritic cell)(Remarks to the Author):

Oberkampff and co-authors demonstrated that cross-presentation ability by pDCs involves endosomal pH alkalization and antigen protection from degradation through a reactive oxygen species (ROS)-dependent mechanism. In contrast to other DC types, however, ROS are not generated by NADPH oxidases, but are exclusively of mitochondrial origin.

This is interesting study suggesting novel mechanism. Technically, it is well prepared. There is only one specific concern regarding measurement of the amount of endocytosed β -lactamase as the indication of cytoplasmic access of antigens. Authors did not clearly explain how they excluded the association of β -lactamase with endosomes and lysosomes in their assay.

However, this study raises several more fundamental concerns.

The main one is that the focus of the study was on unique mechanism of cross-presentation in pDCs in comparison to cDCs. However, there are practically no experiments presented where this pathway was evaluated side-by-side with cDCs isolated from spleens of mice in similar fashion as pDCs. When authors tried to do that (Fig. S5C) they used DCs generated from bone marrow. This is inadequate comparison. BMDCs are different from spleen DCs in many ways. DCs need to be isolated from spleen and, more importantly, there is now enough evidence indicating that CD103⁺ DCs are the major cross-presenting population of DCs, therefore these cells need to be used for comparison. The issue is that cDCs also use Mit ROS and it is not evident that the differences are indeed that strong. Fig.S5C showed decrease in cross-presentation by BMDCs from mitochondrial ROS deficient mice similar to that in pDCs (if one compare different panels of the figures).

Experiments in vivo (Fig. 5) raised additional concern. As presented, it is not possible to distinguish between the role of pDCs and cDCs in induction of CD8⁺ T cells. It is important, since the prevailing thought in the field that conventional DCs are major contributor to cross-presentation dependent immunity. The way how the data are presented one may have an impression that pDCs via their cross-presentation are the major cells in induction of CD8⁺ T cells since mCAT mice had lower T-cell response than NOX2 KO mice. More evidence needs to be presented to justify authors conclusions. In addition, systemic deletion of NOX-2 or neutralization of ROS in mitochondria will have general effect on number cells with important supporting function (cytokine release by macrophages, T cells proliferation, etc). These issues need to be addressed by specific experiments.

Reviewer #2 (DC, antigen presentation)(Remarks to the Author):

Major point:

The authors show in Fig S1B that pDCs are relatively efficient at internalizing apoptotic cells. They did that by co-culturing untreated pDCs with PKH67-labelled apoptotic cells for 30 min, followed by flow cytometric analysis. However, this approach does not conclusively distinguish whether the apoptotic cells are indeed internalized by or alternatively attached to the pDCs. Although, as controls, the authors conduct the study at 4° and 37° C, variations in temperature can also affect cell binding. Besides, labeling of the cell membrane with the dye PKH67 makes the apoptotic cells "stickier". The functional evidence that TLR-activated pDCs that were co-incubated with OVA-loaded apoptotic cells cross-presented OVA-peptides to CD8 T cells (Fig. 1A) is not conclusive evidence that the pDCs internalize OVA-loaded apoptotic cells, since native OVA could leak from the dying cells and then be taken up by the pDCs as a soluble Ag.

Reviewer #3 (mTOR, T metabolism)(Remarks to the Author):

In this paper, the authors investigated the pathways regulating activated pDC cross presentation and proposed that pDC rely on NADPH oxidase-independent ROS production to prime CD8 T cells. The function of ROS in cross presentation is well established. The novelty of this paper is based on the mitochondria derived ROS in regulating cross presentation in activated pDC. However, this conclusion is questionable and not well supported by the experimental results presented in current paper.

1. In Fig 4d, the authors claimed that NOX1/2 deficiency did not affect the ROS production in R848 treated pDC. But the result clearly showed that ROS MFI (about 2500) was reduced compared to WT (about 4500) after R848 treatment. Why? The authors also need to reformat the results to include WT and NOX1/2 KO in the same graph so that the comparison between WT and KO is easier. Same issues are also found in Fig 4 f , g and h, Fig 5 f.

2. In Fig 4h, the authors indicated that there was no cross presentation difference between WT and NOX1/2 KO pDC. The authors need to provide statistical analysis to support their conclusion. In 4mg/ml OVA group, there seems to be differences.

3. The authors concluded that ROS are exclusively of mitochondria origin in pDC, but the evidence is lacking. The authors need to examine total cell ROS in mCAT pDC to prove this point.

4. CpG effects on ROS and cross presentation need to be included.

5. How does R848 treatment induce ROS production in pDC? What is the molecular mechanism? Which gene function is responsible for this? Loss of function or gain of function studies are needed to confirm the role of the candidate genes that the authors proposed in Fig.2.

6. In Fig. 5e and f, antigen degradation is well protected in mCAT DCs while cross presentation is still reduced. What is the reason leading to this cross presentation defect of mCAT pDC?

7. There is only very small reduction of mROS in mCAT pDC (Fig 5d) while the cross presentation of mCAT pDC is dramatically reduced in Fig 5g. What other mechanisms are involved in this process?

8. mROS inhibitors are needed to further validate the requirement of mROS in pDC cross presentation.

9. Do mCAT and NOX1/2 KO CD8 T cells have defects? The system used in Fig 5h cannot exclude this possibility. The authors need to use OT1 transfer system to exclude any potential T cell intrinsic defects in reduced antigen specific CD8 T cell and IFN γ in mCAT mice.

10. Disease model or infection model are needed to highlight the physiological function of ROS mediated pDC cross presentation.

11. Based on Fig 6, the results (surface markers and I-IFN production) got from DPI/NAC treatment and mCAT/NOX1/2 KO genetic model are different. How can the authors use DPI/NAC treatment as evidence to show that ROS production is important for pDC cross presentation?

12. The title contained 'mitochondrial metabolism' but very little data is presented in this aspect. The authors need to revise the title to 'mitochondria-derived ROS' or something similar to accurately reflect the results.

Answers to reviewer comments

We thank the reviewers for their interesting and constructive comments that were very useful to improve our manuscript.

Reviewer #1 (ROS, dendritic cell)

This is interesting study suggesting novel mechanism. Technically, it is well prepared. There is only one specific concern regarding measurement of the amount of endocytosed β -lactamase as the indication of cytoplasmic access of antigens. Authors did not clearly explain how they excluded the association of β -lactamase with endosomes and lysosomes in their assay.

Answer: The FRET reporter has been previously used to measure cytosolic release of model antigens or other molecules into the cytosol (Cebrian et al., Cell, 2011; Segura et al., J Exp Med, 2013, Garcia-Castillo et al., J Cell Sci, 2015). During the measured time-course, the beta-lactamase is indeed within the lumen of the successive endosomal trafficking compartments. However, we have quantified that during the first 90 minutes of uptake, there is no significant degradation of the reporter molecule within the lysosomes (see for example the Figure 6 of Cebrian et al., 2011). Thus, although we do not exclude an association of the reporter molecule with endosomes, we did not observe a reduced enzymatic activity of the β -lactamase, and thus our results do not support the hypothesis that the reporter molecule is degraded within the lysosomes during the selected time-course.

In addition, we have also recently performed additional time-lapse experiments that analyze the transfer of the β -lactamase into the cytosol of different dendritic cells, within the first 60 minutes after addition to the cells. This time-lapse data-piece is part of a manuscript in revision by A Bobard, M Burbage and J Enninga. In this experiment, we used JAWS-II dendritic cells as model, however we got similar data with different DC subsets.

[redacted]

In order to clarify this point, we have introduced a paragraph in the Results section as follows:

Page 10 (1st paragraph): “This approach has already been used to measure cytosolic transfer of antigens in pDCs ¹⁴. During the time course allowing 90 min of β -lactamase internalization, no reduction in the enzymatic activity was observed, showing that the β -lactamase is not processed within lysosomes upon endocytosis before translocation into the cytosol ²⁵.”

However, this study raises several more fundamental concerns.

The main one is that the focus of the study was on unique mechanism of cross-presentation in pDCs in comparison to cDCs. However, there are practically no experiments presented where this pathway was evaluated side-by-side with cDCs isolated from spleens of mice in similar fashion as pDCs. When authors tried to do that (Fig. S5C) they used DCs generated from bone marrow. This is inadequate comparison. BMDCs are different from spleen DCs in many ways. DCs need to be isolated from spleen and, more importantly, there is now enough evidence indicating that CD103⁺ DCs are the major cross-presenting population of DCs, therefore these cells need to be used for comparison.

Answer: We fully agree with this comment that spleen conventional DCs have to be used for comparison with plasmacytoid DCs. In fact, this comparison was presented on the previous Supplementary Figure 5 (corresponding to the new Supplementary Figure 6 of the revised manuscript). However, the legend of this figure was not clearly describing the differences between the experiments done with BMDCs or with spleen DCs.

As mentioned by the reviewer, the panel A of the former Supplementary Figure 5 indeed shows the comparison of the cross-presentation capacity of BMDCs from C57BL/6 and NOX1/2 ^{-/-} mice. We performed this experiment since BMDCs were used in the previous study of Savina et al (Cell, 2006) which demonstrated the involvement of NOX2 in the mechanism of cross-presentation through the control of phagosomal pH. In contrast, our results shown in panels B, C and D of the former Supplementary Figure 5 were obtained with conventional DCs purified from the spleens of C57BL/6, NOX1/2 ^{-/-} and mCAT mice. These cells were obtained after MACS-anti-CD11c labelling and positive selection by Automacs as described in the experimental procedures (page 22 of the revised manuscript). These DCs included CD103⁺ cells which are indeed the major cross-presenting population of DCs. Thus, the former Supplementary Figure 5 B, C and D compares the cross-presenting ability of spleen DCs from the different strains of mice including the CD103⁺ subset.

To clarify this point, we have modified the former Supplementary Figure 5 (Supplementary Figure 6 of the revised manuscript) by adding “BMDC” on panel a and “Spleen cDC” on panels b, c, and d.

We have also modified the legend of the Supplementary Figure 6 as follow:

Page 9 of Supplementary information: “CD11c⁺ cDCs purified from the spleens of C57BL/6 or NOX1/2^{-/-} mice were incubated with serial dilutions of OVA or the SIINFEKL OVA.”

In addition, the Results section has been modified as follows:

Page 11 (1st paragraph): “In contrast, the cross-presentation capacity of BMDCs and splenic cDCs was clearly dependent on NOX (Supplementary Fig. 6a, b) as previously described¹⁹.”

The issue is that cDCs also use Mit ROS and it is not evident that the differences are indeed that strong. Fig.S5C showed decrease in cross-presentation by BMDCs from mitochondrial ROS deficient mice similar to that in pDCs (if one compare different panels of the figures).

Answer: As mentioned in the previous comment, we did not only compare the cross-presentation efficacy of BMDC generated from the different strains of mice, but also of cDCs purified from the spleens of these mice.

To answer the reviewer remark, we have performed a statistical analysis of the cumulative results from two different experiments analyzing the cross-presentation capacity of CD11c⁺ cDCs purified from the spleen of C57BL/6, NOX1/2^{-/-} and mCAT mice. These results clearly demonstrate that the cross-presentation capacity of NOX1/2^{-/-} cDCs was significantly lower than C57BL/6 cDCs and thus, that NOX is involved in the capacity of cDCs to cross-present exogenous Ag as already demonstrated by Savina et al (Cell, 2006). In contrast, this capacity was not affected in mCAT CD11c⁺ cDCs.

These results of this new analysis are shown in the panel d of the revised Supplementary Figure 6. The legend of this figure has been modified as follow:

Page 9 of Supplementary information: “(d) The results show the cumulative data of 2 experiments using cDCs purified from the spleens of C57BL/6, NOX1/2^{-/-} or mCAT mice, loaded with the OVA protein (4 mg/mL) or the SIINFEKL OVA peptide (4 µg/mL) in the absence of DPI. Data are expressed as the mean ± SD percentage of the response obtained with C57BL/6 pDCs.”

Experiments in vivo (Fig. 5) raised additional concern. As presented, it is not possible to distinguish between the role of pDCs and cDCs in induction of CD8⁺ T cells. It is important, since the prevailing thought in the field that conventional DCs are major contributor to cross-presentation dependent immunity. The way how the data are presented one may have an impression that pDCs via their cross-presentation are the major cells in induction of CD8⁺ T cells since mCAT mice had lower T-cell response than NOX2 KO mice. More evidence needs to be presented to justify authors conclusions.

Answer: We fully agree with the reviewer that the results presented on the former Figure 5 do not permit to distinguish between the role of pDCs and cDCs in the induction of CD8⁺ T cells.

To answer this remark and to determine the role of pDCs in these in vivo experiments, we have analyzed the CD8⁺ T cell responses induced by OVA/CpG in the various strains of mice after depletion of pDCs. pDCs were depleted by injections of the InVivoMab antibody (BioXcell), which was specifically designed for the in vivo depletion of pDCs (Kaminsky et al, J. Virol, 2015). This depletion was done before and during the immunization period.

A strong reduction of the OVA-specific T cell responses, assessed by the quantification of the OVA-specific CD8⁺ T cells by the MHC-I-SIINFEKL dextramer and SIINFEKL-specific IFN- γ ELISPOT, was observed in immunized C57BL/6 and NOX1/2^{-/-} mice, which were treated by the anti-CD317 Mab. These results clearly demonstrated that, in this experimental setting, pDCs play a major role in the induction of the OVA-specific CD8⁺ T cell responses. In contrast, the CD8⁺ T cell responses of mCAT mice were not affected by the depletion of pDCs. It is important to note that the low, but still detectable, CD8⁺ T cell responses of anti-CD317 treated C57BL/6 and NOX1/2^{-/-} mice was comparable to the level of responses obtained in mCAT mice treated by the control isotype. The low, but detectable, CD8⁺ T cell responses induced in immunized mCAT mice was not affected by the depletion of pDCs, suggesting that these responses were induced by cDCs.

These new results thus support our conclusions that pDCs are required for the induction of CD8⁺ T cell responses by soluble OVA with CpG as adjuvant and that the lower responses obtained in mCAT mice were due to a defect in the cross-presentation capacity of pDCs.

The results of this experiment are presented on the new Supplementary Figure 9b and the Results section have been modified as follows to clarify this point and to discuss these new results:

Page 12 (3rd paragraph): “We then analyzed the capacity of mCAT mice to develop CD8⁺ T cell responses following in vivo immunization with the OVA soluble antigen with CpG as adjuvant. Indeed, it has been previously established, using Siglec-H deficient mice, that pDCs are required for the in vivo induction of specific CD8⁺ T cell responses by OVA in the presence of CpG³⁰.”

Page 13 (last paragraph): “To confirm the major role of pDCs in the induction of CD8⁺ T cell responses by OVA/CpG, C57BL/6, mCAT and NOX1/2^{-/-} mice were depleted of pDCs by treatment with the anti-CD317 Ab. This depletion of pDCs in C57BL/6 and NOX1/2^{-/-} mice immunized with OVA/CpG was associated with a strong reduction of their capacity to develop OVA-specific T cell responses (Supplementary Fig. 9b), confirming that in this experimental setting, pDCs play a major role in the induction of the OVA-specific CD8⁺ T cell responses. The low, but detectable, CD8⁺ T cell response induced in immunized mCAT mice was not affected by the depletion of pDCs, suggesting that these responses were induced by cDCs.”

In addition, systemic deletion of NOX-2 or neutralization of ROS in mitochondria will have general effect on number cells with important supporting function (cytokine release by macrophages, T cells proliferation, etc). These issues need to be addressed by specific experiments.

Answer: We fully agree with the reviewer that systemic deletion of NOX1/2 or neutralization of ROS in mitochondria could have important effects on the function of different cell populations.

To address this important question, we have performed several experiments to better characterize the immune system of mCAT and NOX1/2^{-/-} mice.

First, we have compared by FACS the immune cell distribution in the spleens of the C57BL/6, NOX1/2^{-/-} and mCAT mice. The results of this new analysis demonstrated that the three strains of mice have a similar immune cell distribution. Although, a lower number of macrophages was observed in NOX1/2^{-/-} mice and a higher number of NK cells was found in mCAT mice, these differences were not statically significant.

Then, to determine whether the innate cells of NOX1/2^{-/-} and mCAT mice respond similarly to in vivo stimulation by CpG, compared to C57BL/6 mice, we have analyzed by the Luminex assay (36-Plex from Affymetrix) the cytokines and chemokines produced in the sera of these mice 9 hours after CpG injection. These results of this experiments showed that the pattern of cytokines and chemokines produced following in vivo stimulation by CpG were similar between these different strains of mice and thus demonstrated that NOX1/2^{-/-} and mCAT mice did not respond differently to in vivo activation by CpG.

Furthermore, to determine if other mechanisms were involved in the lower capacity of pDC from mCAT mice to cross-present antigen following activation, we have compared by the NanoString technology the transcriptomic profile of spleen pDCs purified from either C57BL/6 or mCAT mice, with or without in vitro stimulation with R848. This technology allowed us to compare the expression of about 800 genes. The hierarchical heat map clustering analysis performed demonstrated a similar transcriptional gene profile between pDCs from C57BL/6 and mCAT mice, both at steady state and after R848-stimulation. Furthermore, Multidimensional Dimension Scalling between unstimulated and R848-stimulated pDCs from C57BL/6 and mCAT mice showed that the pattern of genes expressed between these two populations were very closed. Thus, these results strongly suggest that pDCs from C57BL/6 and mCAT mice responded similarly to this stimulation.

Finally, we have grafted tumor cell to mCAT mice and we have followed the tumor growth in these mice, compared to C57BL/6 mice, using two tumor cell models (B16-OVA and TC-1 tumor cells). No difference was observed in the tumor growth control of mCAT and C57BL/6 mice.

These new results clearly establish that NOX1/2^{-/-} and mCAT mice have a functional immune system and fully support the conclusion that the strong reduction of the T cell responses of mCAT in response to immunization with OVA/CpG was not due to an intrinsic defect of the immune cell, but to the lower efficiency of the pDCs to cross-present exogenous Ags.

The results of these new experiments are presented on the new Supplementary Figure 7 panels a and b, Supplementary Figure 8 and Supplementary Figure 10 of the revised manuscript.

The Results and Discussion sections have been modified as follows to discuss these new results:

Page 13 (2nd paragraph): “We then analyzed if this strong alteration of T cell responses of immunized mCAT mice was due to a defect in their immune system. As compared to C57BL/6 and NOX1/2^{-/-} mice, mCAT mice showed a similar immune cell distribution (Supplementary Fig. 7a). In addition, the innate responses of these mice were similar to C57BL/6 and NOX1/2^{-/-} responses, as determined by the pattern of cytokines and chemokines produced following in vivo stimulation by CpG (Supplementary Fig. 7b). The deficiency of mCAT mice in mROS also did not affected their capacity to control tumor growth, as

demonstrated in mice grafted by either B16-OVA or TC-1 tumor cells (Supplementary Fig. 8), confirming that these mice have a functional immune system.”

Page 13 (2nd paragraph): “Thus, these results fully support the conclusion that the strong reduction of the T cell responses of mCAT mice in response to immunization with OVA/CpG was not due to an intrinsic defect of the mCAT CD8⁺ T cells, but to the lower efficiency of the pDCs to cross-present exogenous Ags.”

Page 15 (1st paragraph): “Furthermore, we did not detect major differences in the gene expression of R848- activated pDCs from C57BL/6 and mCAT mice (Supplementary Fig. 10), suggesting that the pDCs from these mice respond similarly to this stimulation”.

Page 15 (2nd paragraph): “Altogether, these data demonstrate that the reduced mROS production of mCAT mice following TLR7 activation affects only the cross-presentation capacity of pDCs, but does not modify other immune functions.”

Page 19 (4th paragraph): “Until now, the studies analyzing the role of pDCs have demonstrated that pDCs are mainly involved in the induction of immune responses through their capacity to produce cytokines and chemokines, and particularly I-IFNs^{30, 44, 45, 46}. However, the present study demonstrates that the strong decrease of CD8⁺ T cell responses observed in mCAT mice following immunization with the OVA soluble antigen was due to a selective defect in their pDCs cross-presentation efficiency, rather than to a general defect of their innate responses and of their capacity to produce cytokines and chemokines.”

In addition to these experiments, since the reviewer 3 suggested that mCAT CD8⁺ T cells could have intrinsic defects, explaining the lower response observed in these mice following OVA immunization, we have analyzed the CD8⁺ T cell responses in the different strains of mice after the transfer of OT-1 CD8⁺ T cells.

5.10⁵ CFSE-labeled purified CD8⁺ OT-I T cells were transferred to C57BL/6, NOX1/2^{-/-} and mCAT mice and, one day later, these mice were immunized with OVA/CpG. The CD8⁺ T cell responses were analyzed four days later by the quantification of the Ag-specific CD8⁺ T cells by the MHC-I-SIINFEKL dextramer and SIINFEKL-specific IFN- γ ELISPOT. A strong reduction of OVA-specific T cell response was observed in transferred mCAT mice, compared to C57BL/6 and NOX1/2^{-/-} mice which have also received OT-1 T cells.

These results fully support the conclusion that the strong reduction of the T cell responses of mCAT mice in response to immunization with OVA/CpG was not due to an intrinsic defect of the mCAT CD8⁺ T cells, but to the lower efficiency of the pDCs to cross-present exogenous Ags.

The results of these new experiments are presented on the new Supplementary Figure 9a of the revised manuscript and the Results section has been modified as follows to discuss these new results:

Page 13 (2nd paragraph): “Finally, a lower OVA-specific T cell response, as compared to C57BL/6 and NOX1/2^{-/-} mice, was observed in mCAT mice after transfer of OVA-specific CD8⁺ OT-I cells and immunization with OVA/CpG (Supplementary Fig. 9a).”

Reviewer #2 (DC, antigen presentation)

Major point:

The authors show in Fig S1B that pDCs are relatively efficient at internalizing apoptotic cells. They did that by co-culturing untreated pDCs with PKH67-labelled apoptotic cells for 30 min, followed by flow cytometric analysis. However, this approach does not conclusively distinguish whether the apoptotic cells are indeed internalized by or alternatively attached to the pDCs. Although, as controls, the authors conduct the study at 4° and 37° C, variations in temperature can also affect cell binding. Besides, labeling of the cell membrane with the dye PKH67 makes the apoptotic cells “stickier”. The functional evidence that TLR-activated pDCs that were co-incubated with OVA-loaded apoptotic cells cross-presented OVA-peptides to CD8 T cells (Fig. 1A) is not conclusive evidence that the pDCs internalize OVA-loaded apoptotic cells, since native OVA could leak from the dying cells and then be taken up by the pDCs as a soluble Ag.

Answer: Our results clearly show that pDCs incubated with Act-mOVA/ K^{b/-} apoptotic cells cross-present OVA. These cells express OVA linked to the cell membrane and thus, we do not think that OVA could leak from the dying cells. Furthermore, these cells could not present directly OVA to OT-I cells since they do not express MHC class I K^b molecule. Indeed, to be presented, OVA should be processed by pDCs, and thus requires the internalization of the apoptotic bodies expressing OVA. Thus, we think that our data support the conclusion that pDCs are able to capture apoptotic cells.

However, we fully agree with the reviewer that the flow cytometric analysis cannot distinguish whether the apoptotic cells are internalized or attached to the cell membrane of pDCs. Thus, to answer this question and to demonstrate that pDCs internalize apoptotic cells, we have incubated pDCs with PKH67-labelled apoptotic cells and then performed a digital imaging analysis using an imaging flow cytometer (ImageStreamX, Amnis corporation).

These new results show that 11% of pDCs were PKH67 positive. Importantly, in these cells, the PKH67 fluorescence was mainly located inside the cells and not at the cell surface, thus demonstrating that pDCs indeed internalized efficiently the dying cells.

The results of this new experiment are presented on the new panel c of the Supplementary Figure 1 and the Results section has been modified as follows:

Page 6 (1st paragraph): “In addition, imaging flow cytometry showed that captured apoptotic cells were located inside the pDCs and not at the cell surface, thus demonstrating that these cells have internalized the dying cells (Supplementary Fig. 1c).”

Reviewer #3 (mTOR, T metabolism)

1. In Fig 4d, the authors claimed that NOX1/2 deficiency did not affect the ROS production in R848 treated pDC. But the result clearly showed that ROS MFI (about 2500) was reduced compared to WT (about 4500) after R848 treatment. Why?

Answer: We fully agree with the reviewer remark and we did not want to conclude that NOX1/2 deficiency did not affect the ROS production in pDCs. However, according to our results showed in the Figure 4b, c, we indeed concluded page 10 that “The R848-activation of pDCs from NOX1^{-/-}, NOX2^{-/-} and NOX1/2^{-/-} mice induced a ROS production that was similar to that in C57BL/6 pDCs (Fig. 4b, c). This production was also inhibited by DPI and NAC (Fig. 4d), demonstrating that the R848 activation induced ROS production independently of NOX1 and 2.”.

In the absence of activation, the MFI for ROS obtained with C57BL/6 pDCs was around 2600 whereas it was around 1800 for NOX1/2^{-/-} pDCs. These results clearly suggest that NADPH oxidases are involved in the basal production of ROS. However, after activation, as shown in the Figure 4b, the increase of the ROS MFI was approximately 1200-1900 for C57BL/6 pDCs and 1100-1300 for NOX1/2^{-/-} pDCs showing that the level of ROS produced after activation is quite similar and that the differences observed between pDCs from C57BL/6 mice and NOX1/2^{-/-} mice were only due to the basal production of ROS in the absence of activation.

These results thus support our conclusion that pDC activation by R848 induces the production of ROS in the absence of NOX1/2.

The authors also need to reformat the results to include WT and NOX1/2 KO in the same graph so that the comparison between WT and KO is easier. Same issues are also found in Fig 4 f, g and h, Fig 5 f.

Answer: The Figures 4 and 5 of the revised manuscript have been reformatted as requested by the reviewer.

In the revised Figure 4, the results showing the inhibition of ROS production by DPI and NAC of pDCs from C57BL/6 and NOX1/2^{-/-} mice are now presented on the same graph (Fig. 4d). We also present the cross-presentation results obtained with NOX1^{-/-} and NOX2^{-/-} mice in the same graph (Fig. 4f), as well as the results of the cross-presentation obtained with the NOX1/2^{-/-} mice (Fig. 4g).

As well, the cross-presentation results obtained with pDCs from C57BL/6 and mCAT mice (former Fig. 5f) are now shown in the same graph and are presented on the new Figure 6d.

2. In Fig 4h, the authors indicated that there was no cross presentation difference between WT and NOX1/2 KO pDC. The authors need to provide statistical analysis to support their conclusion. In 4mg/ml OVA group, there seems to be differences.

Answer: To answer this remark, we have performed a statistical analysis of the cumulative results of 3 different experiments analyzing the cross-presentation ability between R848-activated pDCs purified from C57BL/6 and NOX1/2^{-/-} mice, using OVA at 4 mg/ml. No statistical difference was obtained between the two groups, demonstrating that NOX is not involved in the capacity of pDCs to cross-present exogenous Ag.

We have performed a similar analysis of the cumulative results from two different experiments using CD11c⁺ cDCs purified from the spleens from C57BL/6, NOX1/2^{-/-} and mCAT mice. This analysis demonstrated a significant reduction of the cross-presentation capacity of NOX1/2^{-/-} cDCs compared to C57BL/6 cDCs, confirming that NOX is involved in the capacity of cDCs to cross-present exogenous Ag as demonstrated by Savina et al (Cell, 2006). In contrast, no significant difference was observed with cDCs purified from mCAT mice.

The results of these analysis are presented on the new panel h of the Figure 4 and on the new panel d of the Supplementary Figure 6.

The legends of this Figure have been modified accordingly:

Figure 4 legend (h): “The results show the cumulative data from 3 experiments using R848-activated pDCs from C57BL/6 and NOX1/2^{-/-} mice loaded with the OVA protein (4 mg/mL) or the SIINFEKL OVA peptide (1 µg/mL) and are expressed as the mean ± SD percentage of the response obtained with C57BL/6 pDCs. »

3. The authors concluded that ROS are exclusively of mitochondria origin in pDC, but the evidence is lacking. The authors need to examine total cell ROS in mCAT pDC to prove this point.

Answer: We did not conclude that ROS are exclusively of mitochondria origin in pDC but rather than: “Our observations that DPI and NAC similarly inhibit the production of ROS as assessed by DCFDA and MitoSOX strongly suggest that mROS also constitute the main source of ROS in activated pDCs.”

However, to answer this remark, and as suggested by the reviewer, we have examined the total ROS production in mCAT pDCs activated by R848. pDCs were purified from C57BL/6 and mCAT mice and activated with increasing doses of R848. The ROS production was determined by FACS analysis using DCFDA. As already observed for mROS production (revised Fig. 6a), this experiment demonstrated a significant reduction of total ROS in mCAT pDCs, compared to C57BL/6 pDCs. Although this decrease was statistically different, we did not observe a full inhibition of the ROS production.

These results demonstrate that pDC activation leads to a production of mROS, but that the total ROS produced by the cells following activation were not exclusively of mitochondrial origin.

The results of this experiment are presented in the panel b of the new Figure 6 of the revised manuscript and the Results section has been modified as follows:

Page 12 (1st paragraph): “Indeed, mCAT pDCs produced significantly less mROS compared to C57BL/6 (Fig. 6a), as well as total ROS (Fig. 6b).”

Furthermore, according to these new results, we have deleted the following paragraph in the Discussion of the revised manuscript:

“mROS have been recently demonstrated to be a major source of cellular ROS in macrophages and represent an important component of their antibacterial responses. Our observations that DPI and NAC similarly inhibit the production of ROS as assessed by DCFDA and MitoSOX strongly suggest that mROS also constitute the main source of ROS in activated pDCs.”

4. CpG effects on ROS and cross presentation need to be included.

Answer: As suggested by the reviewer, we analyzed the ROS production and cross-presentation by pDCs following CpG activation.

First, these new results demonstrated that CpG activation induces OVA cross-presentation by pDCs but this activation was less efficient as compared to R848, confirming our previous study (Mouriès et al, Blood, 2008). In addition, we showed that this induction was strongly reduced by the DPI and NAC ROS inhibitors.

We also analyzed, as suggested by the reviewer, the production of total ROS and mROS by purified C57B/6 pDCs following CpG stimulation. The results showed that the activation of pDCs with CpG induced the production of ROS, as demonstrated by FACS analysis with both DCFDA and MitoSox. Thus, as observed for R848, our data demonstrated that CpG induces the production of ROS by mitochondria.

Altogether, these new results demonstrated that both CpG and R848 induce the capacity of pDCs to cross-present Ag and suggest that the induction of cross-presentation by R848 and CpG is regulated by similar mechanism(s).

The results of these experiments are presented in the new Supplementary Figure 4 of the revised manuscript and the Results section has been modified as follows:

Page 7 (last paragraph): “In agreement with our previous study ¹⁶, the capacity of pDCs to cross-present Ags was also induced following CpG activation (Supplementary Fig. 4a, b). This induction was strongly reduced by ROS inhibitors, suggesting that the induction of cross-presentation by R848 and CpG is regulated by similar mechanism(s). Altogether, these data show that ROS are required for the acquisition of the cross-presenting ability by pDCs, following TLR-L activation.”

Page 8 (1st paragraph): “After activation by R848 (Fig. 2a) or CpG (Supplementary Fig. 4c), a strong ROS production was detected.”

Page 11 (2nd paragraph): “After activation by R848 or CpG, a high increase in mROS was observed (Fig. 5a and Supplementary Fig. 4d), which was strongly inhibited by both DPI and NAC and by the superoxide scavenger S3QEL3 (Fig. 5b).

5. *How does R848 treatment induce ROS production in pDC? What is the molecular mechanism? Which gene function is responsible for this? Loss of function or gain of function studies are needed to confirm the role of the candidate genes that the authors proposed in Fig.2.*

Answer: We have carefully analyzed the possibility to realize further investigations on the molecular mechanism(s) involved in the ROS production by pDCs following activation. Our transcriptomic analysis of R848 activated pDCs, using the mouse oxidative stress RT² Profiler PCR array from Qiagen allowing to analyze the expression of genes specifically involved in the regulation of ROS, has suggested that pDC activation regulates the ROS production through ROS metabolizing enzyme as mentioned in the result section (page 8) and discussion (page 17).

To complete this analysis, we have performed a transcriptomic analysis using the nCounter® PanCancer Immune Profiling Panel (NanoString Technologies), which allows the study of approximately 800 genes, not contained in the mouse oxidative stress RT² Profiler PCR array from Qiagen used in our first study. The results obtained did not highlight additional mechanism(s) or pathway(s) potentially involved in the ROS production.

These new results thus suggest that the ROS production by activated pDCs is mainly controlled by the expression of the genes that we have identified using the mouse oxidative stress RT² Profiler PCR array. Further analysis will be required to definitely conclude on the molecular mechanism involved. It will be however difficult to analyze one by one the gene candidates. Indeed, the regulation of ROS involves a large number of genes and the inhibition of the expression of certain genes could be compensated by the over-expression of others.

Thus, although deciphering such regulation will be of great interest, this study will require a large amount of work. Although we fully agree with the interest of this question, we however respectfully consider that the identification of these mechanisms is beyond the scope of the present manuscript.

6. *In Fig. 5e and f, antigen degradation is well protected in mCAT DCs while cross presentation is still reduced. What is the reason leading to this cross presentation defect of mCAT pDC?*

Answer: We agree with the reviewer that antigen degradation is protected in mCAT mice activated pDCs. However, at steady-state, the antigen degradation was more pronounced in mCAT pDCs, compared to C57BL/6 pDCs, in correlation with their lower mROS production following activation. Thus, the antigen protection observed in mCAT pDCs following activation could be due to the residual production of mROS. Furthermore, the percentage of antigen degradation was significantly higher in activated mCAT pDCs, compared to C57BL/6 pDCs. Thus, even if only a small decrease in antigen protection was observed in mCAT pDCs, this difference could have an important effect on the antigen availability and thus, on its efficient processing and presentation.

7. *There is only very small reduction of mROS in mCAT pDC (Fig 5d) while the cross presentation of mCAT pDC is dramatically reduced in Fig 5g. What other mechanisms are involved in this process?*

Answer: We agree with the reviewer that a small reduction of mROS in mCAT pDCs strongly affects the cross-presentation efficacy of pDCs. To answer this remark, we have performed several experiments to better characterize the immune system of mCAT mice, in order to determine whether other mechanisms are involved in this process.

First, we have compared by FACS the immune cell distribution in the spleens of the C57BL/6, NOX1/2^{-/-} and mCAT mice. The results of this new analysis demonstrated that the three strains of mice have a similar immune cell distribution. Although, a lower number of macrophages was observed in NOX1/2^{-/-} mice and a higher number of NK cells was found in mCAT mice, these differences were not statically significant.

Then, to determine whether the innate cells of NOX1/2^{-/-} and mCAT mice respond similarly to in vivo stimulation by CpG, compared to C57BL/6 mice, we analyzed by the Luminex assay (36-Plex from Affymetrix) the cytokines and chemokines produced in the sera of these mice 9 hours after CpG injection. These results of this experiments showed that the pattern of cytokines and chemokines produced following in vivo stimulation by CpG were similar between these different strains of mice and thus demonstrated that NOX1/2^{-/-} and mCAT mice did not respond differently to in vivo activation by CpG.

Furthermore, to determine if other mechanisms were involved in the lower capacity of pDC from mCAT mice to cross-present antigen following activation, we have compared by the NanoString technology the transcriptomic profile of spleen pDCs purified from either C57BL/6 or mCAT mice, with or without in vitro stimulation with R848. This technology allowed us to compare the expression of about 800 genes. The hierarchical heat map clustering analysis performed demonstrated a similar transcriptional gene profile between pDCs from C57BL/6 and mCAT mice, both at steady state and after R848-stimulation. Furthermore, Multidimensional Dimension Scalling between unstimulated and R848-stimulated pDCs from C57BL/6 and mCAT mice showed that the pattern of genes expressed between these two populations were very closed. Thus, these results strongly suggest that pDCs from C57BL/6 and mCAT mice responded similarly to this stimulation.

Finally, we have grafted tumor cell to mCAT mice and we have followed the tumor growth in these mice, compared to C57BL/6 mice, using two tumor cell models (B16-OVA and TC-1 tumor cells). No difference was observed in the tumor growth control of mCAT and C57BL/6 mice.

These new results clearly establish that mCAT mice have a functional immune system and fully support the conclusion that the strong reduction of the T cell responses of mCAT mice in response to immunization with OVA/CpG was not due to an intrinsic defect of the immune cell, but to the lower efficiency of the pDCs to cross-present exogenous Ags.

Furthermore, we also demonstrate that S3QEL2, a specific inhibitor of mROS, totally inhibits the capacity of pDCs to cross-present antigen (see our response to the comment 8), fully supporting our conclusion that the induction of cross-presentation by pDCs is dependent upon mROS. Indeed, even if only a very small decrease of mROS in mCAT pDCs is observed, this reduction strongly correlates with a dramatic inhibition of cross-presentation. We thus can conclude that the main mechanism regulating the induction of cross-presentation is associated with the control of antigen protection, through the production of mROS.

The results of these new experiments are presented on the new Supplementary Figure 7 panels a and b, Supplementary Figure 8 and Supplementary Figure 10 of the revised manuscript.

The Results and Discussion sections have been modified as follows to discuss these new results:

Page 13 (2nd paragraph): “We then analyzed if this strong alteration of T cell responses of immunized mCAT mice was due to a defect in their immune system. As compared to C57BL/6 and NOX1/2^{-/-} mice, mCAT mice showed a similar immune cell distribution (Supplementary Fig. 7a). In addition, the innate responses of these mice were similar to C57BL/6 and NOX1/2^{-/-} responses, as determined by the pattern of cytokines and chemokines produced following in vivo stimulation by CpG (Supplementary Fig. 7b). The deficiency of mCAT mice in mROS also did not affect their capacity to control tumor growth, as demonstrated in mice grafted by either B16-OVA or TC-1 tumor cells (Supplementary Fig. 8), confirming that these mice have a functional immune system.”

Page 13 (2nd paragraph): “Thus, these results fully support the conclusion that the strong reduction of the T cell responses of mCAT mice in response to immunization with OVA/CpG was not due to an intrinsic defect of the mCAT CD8⁺ T cells, but to the lower efficiency of the pDCs to cross-present exogenous Ags.”

Page 15 (1st paragraph): “Furthermore, we did not detect major differences in the gene expression of R848-activated pDCs from C57BL/6 and mCAT mice (Supplementary Fig. 10), suggesting that the pDCs from these mice respond similarly to this stimulation”.

Page 15 (2nd paragraph): “Altogether, these data demonstrate that the reduced mROS production of mCAT mice following TLR7 activation affects only the cross-presentation capacity of pDCs, but does not modify other immune functions.”

Page 19 (4th paragraph): “Until now, the studies analyzing the role of pDCs have demonstrated that pDCs are mainly involved in the induction of immune responses through their capacity to produce cytokines and chemokines, and particularly I-IFNs^{30, 44, 45, 46}. However, the present study demonstrates that the strong decrease of CD8⁺ T cell responses observed in mCAT mice following immunization with the OVA soluble antigen was due to a selective defect in their pDCs cross-presentation efficiency, rather than to a general defect of their innate responses and of their capacity to produce cytokines and chemokines.”

8. mROS inhibitors are needed to further validate the requirement of mROS in pDC cross presentation.

Answer: To address this important question, we have selected the S3QEL3 molecule as a mROS inhibitor. S3QEL3 has been indeed demonstrated to suppress the superoxide production by the complex III in the mitochondria without affecting the normal electron flux or cellular oxidative phosphorylation.

We have tested by FACS using Mitosox the production of mROS by purified pDCs following R848 activation in the presence of S3QEL3. These results demonstrated that S3QEL3 inhibits the mROS production by pDCs following activation. In addition, S3QEL3 strongly inhibited the cross-presentation capacity of activated pDCs. Altogether, these results confirm that the

induction of cross-presentation in pDCs activated by R848 is dependent upon mROS production.

The results of these new experiments are presented on the new Figure 5, panels b, c of the revised manuscript and the Results section has been modified as follows:

Page 11 (last paragraph): “After activation by R848 or CpG, a high increase in mROS was observed (Fig. 5a and Supplementary Fig. 4d), which was strongly inhibited by both DPI and NAC and by the superoxide scavenger S3QEL3 (Fig. 5b). S3QEL3 has been demonstrated to suppress superoxide production by the complex III in the mitochondria without affecting normal electron flux or cellular oxidative phosphorylation²⁸. Importantly, S3QEL3 also inhibited the cross-presentation capacity of pDCs induced by R848 (Fig. 5c, d), showing its dependency upon mROS production.”

Page 17 (1st paragraph): “The production of mROS by the R848-activated pDCs was strongly reduced in the presence of the NAC and S3QEL2 which are ROS and mROS scavengers, respectively, but also by DPI, which has been mainly described as a specific inhibitor of flavoenzymes, such as NOX2, but is also a potent inhibitor of mitochondrial complex I in the respiratory chain⁴⁰.”

9. Do mCAT and NOX1/2 KO CD8 T cells have defects? The system used in Fig 5h cannot exclude this possibility. The authors need to use OT1 transfer system to exclude any potential T cell intrinsic defects in reduced antigen specific CD8 T cell and IFN γ in mCAT mice.

Answer: We agree with the reviewer that CD8⁺ T cells from mCAT could have intrinsic defects, which could be responsible for the lower CD8⁺ T cell responses observed in these mice following OVA immunization.

To address this question, and as suggested by the reviewer, we transferred 5.10⁵ CFSE-labeled purified CD8⁺ OT-I T cells to C57BL/6, NOX1/2^{-/-} and mCAT mice and, one day later, these mice were immunized with OVA/CpG. The CD8⁺ T cell responses were analyzed four days later by the quantification of the Ag-specific CD8⁺ T cells by the MHC-I-SIINFEKL dextramer and SIINFEKL-specific IFN- γ ELISPOT. A strong reduction of OVA-specific T cell responses was observed in transferred mCAT mice, compared to C57BL/6 and NOX1/2^{-/-} mice, which have also received OT-1 T cells.

These results fully support the conclusion that the strong reduction of the T cell responses of mCAT mice in response to immunization with OVA/CpG was not due to an intrinsic defect of the mCAT CD8⁺ T cells, but to the lower efficiency of the pDCs to cross-present exogenous Ags.

The results of these new experiments are presented on the new Supplementary Figure 9a of the revised manuscript and the Results section has been modified as follows to discuss these new results:

Page 13 (2nd paragraph): “Finally, a lower OVA-specific T cell response, as compared to C57BL/6 and NOX1/2^{-/-} mice, was observed in mCAT mice after transfer of OVA-specific CD8⁺ OT-I cells and immunization with OVA/CpG (Supplementary Fig. 9a).”

10. Disease model or infection model are needed to highlight the physiological function of ROS mediated pDC cross presentation.

Answer: To select such disease or infection model, we had first to better characterize the mCAT mice, which represents an appropriate model to address to the physiological function of ROS mediated pDC cross presentation. The new experiments performed for the revision of the manuscript clearly established that mCAT mice have a functional immune system and fully support the conclusion that the strong reduction of the T cell responses of mCAT in response to immunization with OVA/CpG was not due to an intrinsic defect of the immune cell, but to the lower efficiency of the pDCs to cross-present exogenous Ags. In particular, we have demonstrated the capacity of mCAT mice to produce cytokines or chemokines in response to CpG and to control tumor growth.

Our results thus support the hypothesis that a defect in mROS production, as observed with the mCAT mice, affects selectively the capacity of pDCs to cross-present exogenous Ags and thus would impact only the induction of CD8⁺ T cell responses by pDCs. To our knowledge, so far, there is no disease or infection model which is solely controlled by CD8⁺ T cell responses induced by pDCs. At this stage, it thus difficult to evidence the physiological function of ROS mediated pDC cross-presentation using such models. Indeed, no difference was observed in the control of tumor growth by mCAT and C57BL/6 mice, using two different tumor models (B16-OVA and TC-1 tumor cells).

However, our study demonstrates that a defect in mROS production greatly impacts the induction of CD8⁺ T cell responses following immunization using CpG as an adjuvant. Thus, these results highlight a crucial role of mROS that could be essential for the development of new immunotherapeutic strategies targeting pDCs. In a near future, we will compare in mCAT and C57BL/6 mice the therapeutic efficacy of vaccination with tumor antigen and CpG against the growth of grafted tumor cells expressing this antigen. If our hypothesis is correct, such therapeutic treatment should have a reduced efficacy in mCAT mice, demonstrating the physiological relevance of our findings.

11. Based on Fig 6, the results (surface markers and I-IFN production) got from DPI/NAC treatment and mCAT/NOX1/2 KO genetic model are different. How can the authors use DPI/NAC treatment as evidence to show that ROS production is important for pDC cross presentation?

Answer: We fully agree with the reviewer that the results (surface markers and I-IFN production) obtained from DPI/NAC treatment and mCAT/NOX1/2 KO genetic model are different.

However, DPI and NAC treatment were used to demonstrate that the ROS production is important for the antigen cross-presentation by pDCs, since these inhibitors did not affect the presentation of the synthetic SIINFEKL peptide to OT-I T cells (Fig. 1b and Supplementary Fig. 2b). This result demonstrates that these inhibitors did not affect the activation of T cells when the antigen is already processed. In contrast, in these experiments, the presentation of the soluble OVA protein was totally inhibited by NAC and DPI (Fig. 1b, c). DPI also inhibited the presentation of particulate antigens, such as beads and apoptotic cells to T cells (Supplementary Fig. 2b).

It is also important to mention that DPI and NAC inhibitors were used in this study to confirm that the inhibition of ROS production fully abolishes exogenous antigen presentation in a cross-presentation assay, as previously demonstrated for cDCs (Savina et al, Cell 2009). The specificity of these inhibitors could however be questioned, in particular for DPI, which is an inhibitor of flavoenzymes or for NAC, which is an ROS scavenger. Thus, DPI and NAC could potentially inhibit other sources of ROS than NOX1/2 or mitochondria. Thus, one can speculate that their effect could be less selective, or more marked, than the genetic models.

In contrast, in the mCAT mouse model, the reduction of the production of mROS by mitochondria is highly selective. Since it affects only the induction of cross-presentation by pDCs and of CD8⁺ T cell responses in vivo, in the absence of alteration of the expression of costimulatory molecules nor of IFN- α production, we can indeed conclude that ROS production is important for pDC cross presentation.

12. The title contained 'mitochondrial metabolism' but very little data is presented in this aspect. The authors need to revise the title to 'mitochondria-derived ROS' or something similar to accurately reflect the results.

Answer: The title of the revised manuscript has been revised following the recommendation of the reviewer as follows: “Mitochondrial reactive oxygen species regulate the induction of CD8⁺ T cell responses by plasmacytoid dendritic cells”.

REVIEWERS' COMMENTS:

Reviewer #1 (Remarks to the Author):

Authors addressed all my concerns.

Reviewer #2 (Remarks to the Author):

The authors have addressed my concerns properly.

Reviewer #3 (Remarks to the Author):

The authors did a clearly thorough job in addressing my previous questions, and the manuscript is considerably improved. I have two follow-up minor comments.

For my original critique #5, the authors did additional exploratory approaches. Although the data are largely negative, they need to be included in the manuscript to highlight the significance of their positive findings.

For my critique #10, I understand that the authors did not find phenotypes in certain in vivo models examined. Such negative results need to be presented and discussed in the paper. It is important to give readers a fair understanding of the model and the physiological impact.

Answers to reviewer comments

Reviewer #1:

Authors addressed all my concerns.

Reviewer #2:

The authors have addressed my concerns properly.

Answer: We thank both reviewers for their positive comments.

Reviewer #3:

The authors did a clearly thorough job in addressing my previous questions, and the manuscript is considerably improved.

Answer: We thank the reviewer for these positive comments.

I have two follow-up minor comments. For my original critique #5, the authors did additional exploratory approaches. Although the data are largely negative, they need to be included in the manuscript to highlight the significance of their positive findings.

Answer: We have indeed included these data in the revised manuscript but we agree that our answer was not precise enough.

Indeed, the original critique #5 was: *How does R848 treatment induce ROS production in pDC? What is the molecular mechanism? Which gene function is responsible for this? Loss of function or gain of function studies are needed to confirm the role of the candidate genes that the authors proposed in Fig.2.*

Our previous answer was: We have carefully analyzed the possibility to realize further investigations on the molecular mechanism(s) involved in the ROS production by pDCs following activation.

Our transcriptomic analysis of R848 activated pDCs, using the mouse oxidative stress RT² Profiler PCR array from Qiagen allowing to analyze the expression of genes specifically involved in the regulation of ROS, has suggested that pDC activation regulates the ROS production through ROS metabolizing enzyme as mentioned in the result section (page 8) and discussion (page 17).

To complete this analysis, we have performed a transcriptomic analysis using the nCounter® PanCancer Immune Profiling Panel (NanoString Technologies), which allows the study of approximately 800 genes, not contained in the mouse oxidative stress RT² Profiler PCR array from Qiagen used in our first study. The results obtained did not highlight additional mechanism(s) or pathway(s) potentially involved in the ROS production.

These new results thus suggest that the ROS production by activated pDCs is mainly controlled by the expression of the genes that we have identified using the mouse oxidative stress RT² Profiler PCR array. was not enough precise.

Complementary answer:

The results of these new experiments are presented on the new Supplementary Figure 10 of the revised manuscript.

For my critique #10, I understand that the authors did not find phenotypes in certain in vivo models examined. Such negative results need to be presented and discussed in the paper. It is important to give readers a fair understanding of the model and the physiological impact.

Answer: Similarly, we have indeed included these data in the revised manuscript but we agree that our answer was not precise enough.

Indeed, the original critique #10 was: *Disease model or infection model are needed to highlight the physiological function of ROS mediated pDC cross presentation.*

Our previous answer was: To select such disease or infection model, we had first to better characterize the mCAT mice, which represents an appropriate model to address to the physiological function of ROS mediated pDC cross presentation. The new experiments performed for the revision of the manuscript clearly established that mCAT mice have a functional immune system and fully support the conclusion that the strong reduction of the T cell responses of mCAT in response to immunization with OVA/CpG was not due to an intrinsic defect of the immune cell, but to the lower efficiency of the pDCs to cross-present exogenous Ags. In particular, we have demonstrated the capacity of mCAT mice to produce cytokines or chemokines in response to CpG and to control tumor growth.

Our results thus support the hypothesis that a defect in mROS production, as observed with the mCAT mice, affects selectively the capacity of pDCs to cross-present exogenous Ags and thus would impact only the induction of CD8⁺ T cell responses by pDCs. To our knowledge, so far, there is no disease or infection model which is solely controlled by CD8⁺ T cell responses induced by pDCs. At this stage, it thus difficult to evidence the physiological function of ROS mediated pDC cross-presentation using such models. Indeed, no difference was observed in the control of tumor growth by mCAT and C57BL/6 mice, using two different tumor models (B16-OVA and TC-1 tumor cells).

Complementary answer:

The results of these new experiments are presented on the new Supplementary Figures 7 and 8 of the revised manuscript.

The Results and Discussion sections have been modified as follows to discuss these new results:

Page 13 (2nd paragraph): “We then analyzed if this strong alteration of T cell responses of immunized mCAT mice was due to a defect in their immune system. As compared to C57BL/6 and NOX1/2^{-/-} mice, mCAT mice showed a similar immune cell distribution

(Supplementary Fig. 7a). In addition, the innate responses of these mice were similar to C57BL/6 and NOX1/2^{-/-} responses, as determined by the pattern of cytokines and chemokines produced following in vivo stimulation by CpG (Supplementary Fig. 7b). The deficiency of mCAT mice in mROS also did not affected their capacity to control tumor growth, as demonstrated in mice grafted by either B16-OVA or TC-1 tumor cells (Supplementary Fig. 8), confirming that these mice have a functional immune system.”